# MetaMask: Revisiting Dimensional Confounder for Self-Supervised Learning

**Jiangmeng Li**[*], **Wenwen Qiang**[*], **Yanan Zhang**
University of Chinese Academy of Sciences
Institute of Software Chinese Academy of Sciences
Southern Marine Science and Engineering Guangdong Laboratory (Guangzhou)
{jiangmeng2019, wenwen2018, yanan2018}@iscas.ac.cn

**Wenyi Mo**
Gaoling School of Artificial Intelligence
Renmin University of China
2022101010@ruc.edu.cn

**Changwen Zheng**
Institute of Software Chinese Academy of Sciences
Southern Marine Science and Engineering Guangdong Laboratory (Guangzhou)
changwen@iscas.ac.cn

**Bing Su**[†]
Gaoling School of Artificial Intelligence
Renmin University of China
Beijing Key Laboratory of Big Data Management and Analysis Methods
subingats@gmail.com

**Hui Xiong**
Thrust of Artificial Intelligence
The Hong Kong University of Science and Technology (Guangzhou)
Guangzhou HKUST Fok Ying Tung Research Institute
xionghui@ust.hk

## Abstract

As a successful approach to self-supervised learning, contrastive learning aims to learn invariant information shared among distortions of the input sample. While contrastive learning has yielded continuous advancements in sampling strategy and architecture design, it still remains two persistent defects: the interference of task-irrelevant information and sample inefficiency, which are related to the recurring existence of trivial constant solutions. From the perspective of dimensional analysis, we find out that the *dimensional redundancy* and *dimensional confounder* are the intrinsic issues behind the phenomena, and provide experimental evidence to support our viewpoint. We further propose a simple yet effective approach *MetaMask*, short for the *dimensional **Mask** learned by **Meta**-learning*, to learn representations against dimensional redundancy and confounder. MetaMask adopts the redundancy-reduction technique to tackle the dimensional redundancy issue

---

[*]Equal contributions.
[†]Corresponding author.

36th Conference on Neural Information Processing Systems (NeurIPS 2022).

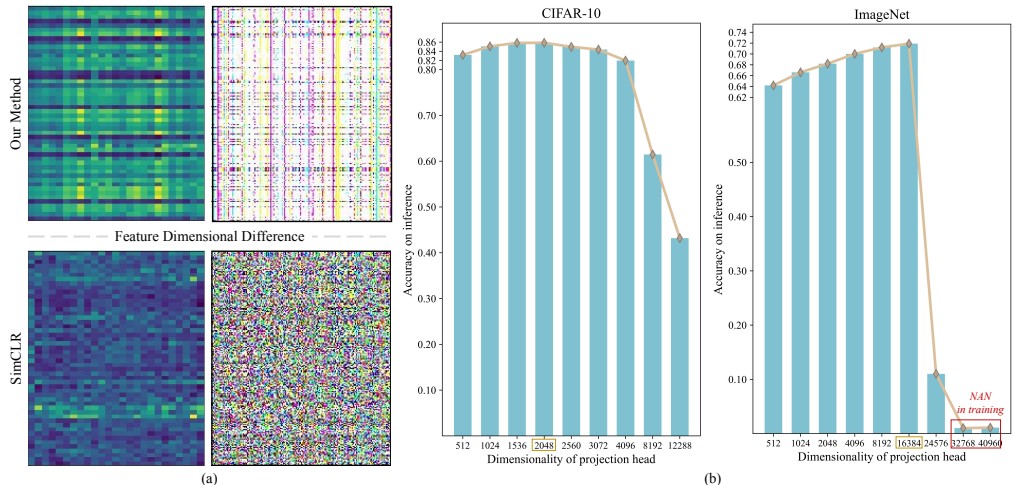

Figure 1: (a) The visualization of the representations learned by SimCLR [1] and our method using the redundancy-reduction technique [2] on CIFAR-10, respectively. The learned features are projected into a color image in RGB format, where different colors represent different types of features. The abscissa axis represents the feature dimensions, and the ordinate axis represents samples of different classes. The greater the color contrast, the lower the dimensional feature similarity. The left plots present the contribution of different dimensions to a specific category classification, and the right plots present the similarity between dimension features within a batch. Compared with SimCLR, our method using redundancy-reduction can indeed learn representations with decoupled dimensions. (b) The experimental results obtained by Barlow Twins on CIFAR-10 and ImageNet datasets. We intuitively increase the dimensionality of the projection head, which leads the dimensions of representations to model more different information. Note that we adopt the official code of Barlow Twins and train on 8 GPUs of NVIDIA Tesla V100.

and innovatively introduces a dimensional mask to reduce the gradient effects of specific dimensions containing the confounder, which is trained by employing a meta-learning paradigm with the objective of improving the performance of masked representations on a typical self-supervised task. We provide solid theoretical analyses to prove MetaMask can obtain tighter risk bounds for downstream classification compared to typical contrastive methods. Empirically, our method achieves state-of-the-art performance on various benchmarks.[1]

# 1   Introduction

A fundamental idea behind self-supervised learning is to learn discriminative representations from the input data without relying on human annotations. Recent advances in visual self-supervised learning [3, 4, 5, 6, 7] demonstrate that unsupervised approaches can achieve competitive performance over supervised approaches by introducing sophisticated self-supervised tasks. A representative learning paradigm is contrastive learning [8, 1, 9, 10, 11, 12, 13], which aims to learn invariant information from different views (generated by data augmentations) by performing instance-level contrast, i.e., pulling views of the same sample together while pushing views of different samples away. Various[14, 15, 16, 17] techniques have been explored to address the problem of trivial solutions to self-supervised contrastive learning, e.g., constant representations. However, state-of-the-art methods still suffer from two crucial defects, including the interference of task-irrelevant information and sample inefficiency. The trivial solution problem is related to such defects, e.g., a small number of negative samples or a large proportion of task-irrelevant information may lead the model to learn constant representations. We revisit the current self-supervised learning paradigm from the perspective of dimensional analysis and argue that the intrinsic issues behind the defects of self-supervised learning are the *dimensional redundancy* and *dimensional confounder* [2].

---

[1]The implementation is available at `https://github.com/lionellee9089/MetaMask`

[2]Throughout this paper, the term *confounder* is used in its idiomatic sense rather than the specific statistical sense in Structural Causal Models [18].

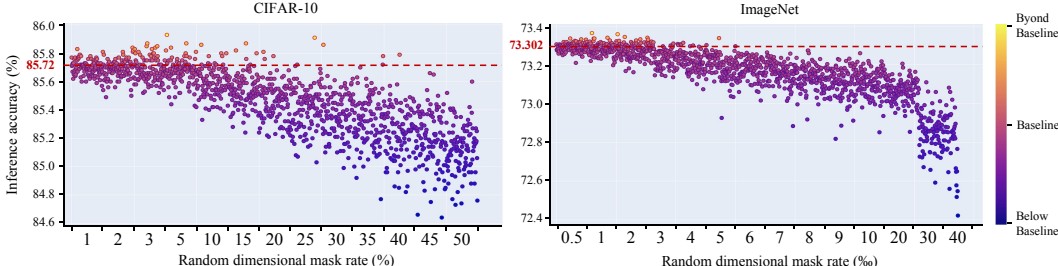

Figure 2: Experimental scatter diagrams obtained by Barlow Twins [2] with randomly masked dimensions on CIFAR-10 and ImageNet datasets, where *Baseline* and the *red dashed lines* denote the performance achieved by the unmasked representation of Barlow Twins. Every single point denotes an independent experimental result achieved by randomly masking the original representation at the dimensional level with a specific mask rate. Note that the original representation is consistently fixed. Following the experimental protocol of [2, 16], we evaluate the performance on CIFAR-10 and ImageNet by adopting KNN prediction and linear probing, respectively, which further proves the general existence of dimensional confounders.

From the perspective of information theory, each dimension holds a subset of the representation's information entropy. The dimensional redundancy denotes that multiple dimensions hold overlapping information entropy. To tackle this issue, Barlow Twins [2] takes advantage of a neuroscience approach, i.e., redundancy-reduction [19], in self-supervised learning. It makes the cross-correlation matrix of twin embeddings close to the identity matrix, which is conceptually simple yet effective to address the dimensional redundancy issue. Figure 1 (a) supports the superiority of redundancy-reduction in addressing the dimensional redundancy problem. Since Barlow Twins encourages the dimensions of learned representations to model decoupled information, it naturally avoids the collapsed trivial solution that outputs the same vector for all images.

However, only reducing dimensional redundancy still suffers from the problem of dimensional confounder. The dimensional confounder indicates a set of dimensions that contains "harmful" information and further degenerates the performance of the model, e.g., the dimensions simply capturing chaotic and worthless background information from the input. To prove the existence of such dimensional confounders, we conduct motivating experiments from two perspectives. It is stated in [2] that Barlow Twins keeps improving as the dimensionality of the projection head increases, since the backbone network acts as a dimensionality bottleneck to constrain the dimensionality of the representation so that the learning paradigm of Barlow Twins can promote the representation capture more task-irrelevant information with the same dimensionality. Counterintuitively, our observation from the motivating experiments is in stark contrast with the conclusion in [2]. As shown in Figure 1 (b), the performance of Barlow twins as a function of the dimensionality presents a convex pattern on either CIFAR-10 or ImageNet. After reaching the peak, the performance gradually decreases as the dimensionality increases until it collapses. We reckon that since the task-relevant semantic information is limited, as the information captured in different dimensions is gradually decoupled, more task-irrelevant noisy information is encoded into the representation, i.e., the dimensional confounder increases, so the performance of the learned representation is getting worse until collapse.

To further validate our inference, we randomly mask several dimensions by frankly setting the masked values to 0, and then we evaluate the performance of the masked representation. The experimental results are demonstrated in Figure 2, where every single point denotes the result of a sampled mask. The representations with some specific masked dimensions can indeed achieve better performance than the unmasked representation, and such an observation is consistent on a small-scaled dataset, e.g., CIFAR-10, and a large-scale dataset, e.g., ImageNet. The dimensional mask rate that can improve the performance of the model on ImageNet is on average lower than that on CIFAR-10. We understand that this phenomenon depends on an ingredient: the representation learned on CIFAR-10 models discriminative information for 10 categories in 512 dimensions, while the representation uses 2048 dimensions to model information for 1000 categories on ImageNet, so the dimension-to-class ratio on CIFAR-10 (51.2) is much higher than that on ImageNet (2.048). Therefore, the presence of dimensional confounders naturally remains at a low level on ImageNet since supporting 1000-category classification requires a large amount of heterogeneous discriminative information. This empirical evidence further supports our assumption of the dimensional confounder.

To this end, we intuitively propose a simple yet effective approach *MetaMask*, short for the ***Meta-learned dimensional Mask***, to adjust the impacts of different dimensions in a self-paced manner. In order to learn representations against dimensional redundancy and confounder, we regard the cross-correlation matrix constraint of Barlow Twins as a dimensional regularization term and further introduce a dimensional mask, which simultaneously enhances the gradient effect of the dimensions containing discriminative information and reduces that of the dimensions containing the confounder during training. We employ the meta-learning paradigm to train the learnable dimensional mask with the optimization objective of improving the performance of masked representations on typical self-supervised tasks, e.g., contrastive learning. Consequently, the dimensional mask incessantly adjusts the weights of different dimensions with respect to their gradient contribution to the optimization, so as to make the model focus on the acquisition of discriminative information, which can address the trivial solution caused by treating all dimensions equally in optimization, especially for high-dimensional representations. Empirically, we conduct head-to-head comparisons on various benchmark datasets, which prove the effectiveness of MetaMask. Concretely, the **contributions** of this paper are four-fold:

- We present a timely study on the adaptation of the state-of-the-art self-supervised models on downstream applications and identify two critical issues associated with the learned representation, i.e., dimensional redundancy and dimensional confounder.

- We propose a novel approach to tune the gradient effects of dimensions in a self-paced manner during optimization according to their improvements on a self-supervised task.

- Theoretically, we provide theorems and proofs to support that MetaMask achieves relatively tighter risk bounds for downstream classification compared to typical contrastive methods.

- We conduct extensive experiments to demonstrate the superiority of MetaMask over benchmark methods on downstream tasks.

## 2   Related Work

**Self-supervised learning** achieved impressive success in the field of representation learning without human annotation by constructing auxiliary self-supervised tasks. Typically, self-supervised approaches can be categorized as either generative or discriminative [20, 1, 14]. Conventional generative approaches bridge the input data and latent embedding by building a distribution over them, and the learned embeddings are treated as image representations [21, 22, 23, 24, 25, 26]. However, such a learning paradigm learns the distribution at the pixel-level, which is computationally expensive. The detailed information required for image generation may be redundant for representation learning.

For discriminative methods, contrastive learning achieves state-of-the-art performance on various downstream tasks [6, 27, 28, 29, 30, 31, 32, 33, 15]. Deep InfoMax [3] explores to maximize the mutual information between the input and output of an encoder. CPC [5] adopts noise-contrastive estimation [34] as the contrastive loss to train the model by maximizing the mutual information of multiple views, which is deduced by using the Kullback-Leibler divergence [35]. CMC [8] and AMDIM [4] employs contrastive learning on the multi-view data. SimCLR [1] and MoCo [9, 36] use large batch or memory bank to enlarge the amount of available negative features. To sample informative negatives, DebiasedCL[10] and HardCL [11] design sampling strategies by using positive-unlabeled learning methods [37, 38]. Motivated by [39], [40] provides an information theoretical framework for self-supervised learning and proposes an information bottleneck-based approach. However, these methods rely on a large number of negative samples to learn valuable representations while avoiding trivial solutions, i.e., constant representation. SimSiam [16] and BYOL [14] jointly waive negatives and avoid training degradation by using the techniques of asymmetric siamese network and stop-gradient. SSL-HSIC [41] provides a unified framework for negative-free contrastive learning based on Hilbert Schmidt Independence Criterion [42]. The representation learning paradigm employed by current contrastive methods is contrary to the coding criteria proposed by [19] and [43], respectively. Therefore, Barlow Twins [2] proposes to constrain the cross-correlation matrix between the features to be close to the identity matrix, which can effectively reduce the dimensional redundancy. Yet, such an approach still suffers from the dimensional confounder issue, and we thus propose a new meta-learning-based method to attenuate the over-learning of dimensional confounders by learning a mask to filter out irrelevant information.

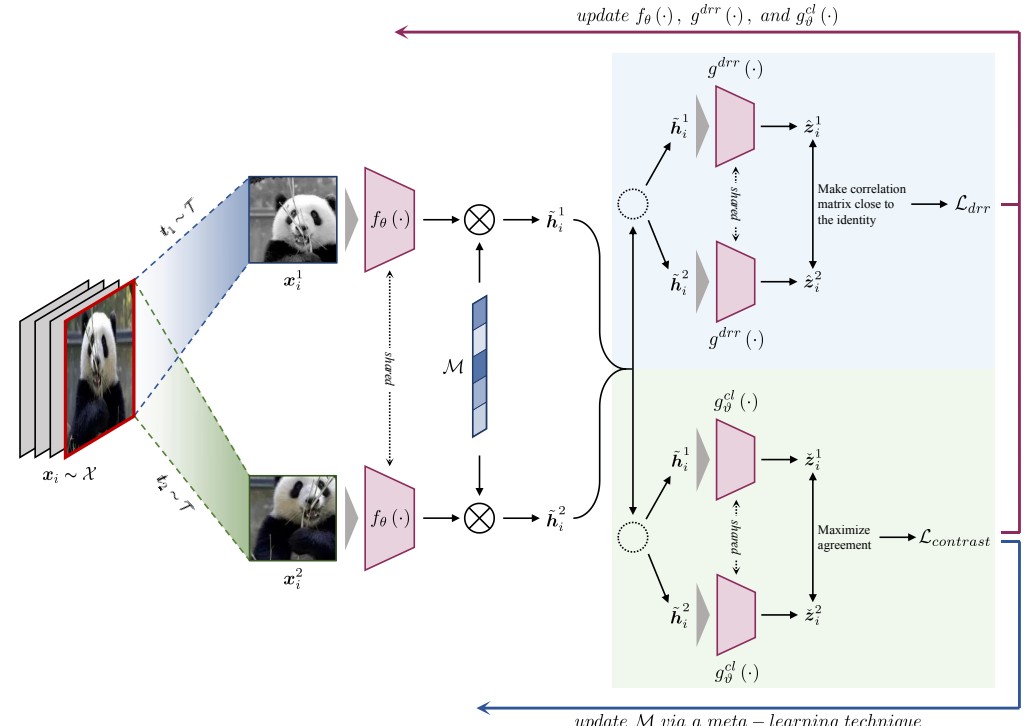

Figure 3: The architecture of MetaMask. The solid red line pointing backwards represents the regular training step, where the encoder and projection heads are trained by jointly back-propagating $\mathcal{L}_{drr}$ and $\mathcal{L}_{contrast}$. The solid blue line pointing backwards represents the meta-learning step, where the dimensional mask $\mathcal{M}$ is trained by performing the second-derivative technique on $\mathcal{L}_{contrast}$. All learnable networks are trained until convergence.

## 3 Preliminaries

Formally, we suppose the input multi-view dataset as $\mathcal{X} = \left\{ \boldsymbol{x}_i^j \middle| i \in [\![1, N^*]\!], j \in [\![1, M]\!] \right\}$, where $N^*$ denotes the number of samples, and $M$ denotes the number of views. Therefore, $\boldsymbol{x}_i^j$ represents the $j$-th view of the $i$-th sample following $\boldsymbol{x}_i^j \sim \mathcal{X}$. The multiple views are built by performing identically distributed data augmentations [8, 1, 2], i.e., $\{\boldsymbol{t}_j\}_{j=1}^M \sim \mathcal{T}$.

### 3.1 Contrastive Learning

We recap the preliminaries of contrastive learning [1], which aims to learn an embedding that jointly maximizes agreement between views of the same sample and separates the views of different samples in the latent space. Given a minibatch of $N$ examples sampled from the multi-view dataset $\mathcal{X}$, *positives* represent the pairs $\{\boldsymbol{x}_i^j, \boldsymbol{x}_i^{j'}\}$ where $j, j' \in [\![1, M]\!]$, and *negatives* represent $\{\boldsymbol{x}_i^j, \boldsymbol{x}_{i'}^{j'}\}$ where $i \neq i'$. Conventional contrastive methods feed the input $\boldsymbol{x}_i^j$ into a view-shared encoder $f_\theta(\cdot)$ to learn the representation $\boldsymbol{h}_i^j$, which is mapped into a feature $\boldsymbol{z}_i^j$ by a projection head $g_\vartheta(\cdot)$, where $\theta$ and $\vartheta$ are the network parameters. $f_\theta(\cdot)$ and $g_\vartheta(\cdot)$ are trained by using the contrastive loss [5, 1]:

$$\mathcal{L}_{contrast} = - \sum_{\substack{i \in [\![1, N]\!] \\ j \in [\![1, M-1]\!] \\ j^+ \in [\![j+1, M]\!]}} \log \frac{\exp\left[d\left(\boldsymbol{z}_i^j, \boldsymbol{z}_i^{j^+}\right)/\tau\right]}{\sum_{i' \in [\![1, N]\!]} \sum_{j' \in [\![1, M]\!]} \mathbb{1}_{[i \neq i' \vee j \neq j']} \cdot \exp\left[d\left(\boldsymbol{z}_i^j, \boldsymbol{z}_{i'}^{j'}\right)/\tau\right]}$$

(1)

where $d(\cdot)$ is a discriminating function to measure the similarity of the feature pairs, $\mathbb{1}_{[i \neq i' \vee j \neq j']}$ denotes an indicator function equalling to 1 if $i \neq i'$ or $j \neq j'$, and $\tau$ is a temperature parameter valued by following [1]. In inference, the projection head $g_\vartheta(\cdot)$ is discarded, and the representation $\boldsymbol{h}_i^j$ is directly used for downstream tasks.

## 3.2 Dimensional Redundant Reduction

Following the principle proposed by [19], [2] introduces a redundancy-reduction objective function:

$$\mathcal{L}_{drr} = \sum_{\substack{j \in [\![1,M-1]\!] \\ j^+ \in [\![j+1,M]\!]}} \left[ \sum_{k \in [\![1,D]\!]} \left( 1 - C_{kk}^{j,j^+} \right)^2 + \lambda \sum_{\substack{k,k' \in [\![1,D]\!] \\ k \neq k'}} \left( C_{kk'}^{j,j^+} \right)^2 \right] \tag{2}$$

where $\lambda$ is a positive hyper-parameter valued by following [2], and $C$ is the cross-correlation matrix computed between the outputs of the encoder along the batch dimension:

$$C_{kk'}^{j,j^+} = \frac{\sum_{i \in [\![1,N]\!]} z_{i,k}^j \cdot z_{i,k'}^{j^+}}{\sqrt{\sum_{i \in [\![1,N]\!]} \left( z_{i,k}^j \right)^2} \cdot \sqrt{\sum_{i \in [\![1,N]\!]} \left( z_{i,k'}^{j^+} \right)^2}} \tag{3}$$

where $k$ and $k'$ index the dimension of features. $C$ is a square matrix with the size of $D$, which denotes the dimensionality of the projected feature $z$. The value range of the elements in $C$ is $[-1, 1]$, i.e., complete anti-correlation to complete correlation.

## 4 Methodology

Our goal is to learn discriminative representations against dimensional redundancy and dimensional confounder from multiple views by performing self-supervised learning. To this end, we propose MetaMask to automatically determine the optimal dimensionality of the representation by decoupling information into different dimensions and filtering out those dimensions with irrelevant information. MetaMask addresses the dimensional redundancy issue by adopting a redundancy-reduction regularization. To tackle the dimensional confounder problem, MetaMask learns a dimensional mask to adjust the gradient weights of dimensions according to their contributions to improving a specific self-supervised task during optimization, which is achieved in a meta-learning manner.

MetaMask's architecture is shown in Figure 3. Specifically, the sample $x_i^j$ is first fed into the encoder to obtain its representation $h_i^j$. Then, we build a learnable dimensional mask $\mathcal{M} = \left\{ \omega_k | k \in [\![1,D]\!] \right\}$, which is introduced to assign a *weight* to each dimension of the representation $h_i^j$ by

$$\tilde{h}_i^j = h_i^j \otimes \mathcal{M} \tag{4}$$

where $\tilde{h}_i^j$ denotes the masked representation, and $\cdot \otimes \cdot$ is an element-wise Hadamard product function. $\tilde{h}_i^j$ is then fed into two separate projection heads $g^{drr}(\cdot)$ and $g_\vartheta^{cl}(\cdot)$, respectively. $g_\vartheta^{cl}(\cdot)$ plays the same role as in conventional contrastive learning methods, i.e., the contrastive loss $\mathcal{L}_{contrast}$ in Eq. (1) is applied to the output features of $g_\vartheta^{cl}(\cdot)$. For the projection head $g^{drr}(\cdot)$, the redundancy-reduction loss $\mathcal{L}_{drr}$ in Eq. (2) is performed to its output features to tackle the dimensional redundancy.

In training, the encoder $f_\theta(\cdot)$, the projection heads $g^{drr}(\cdot)$ and $g_\vartheta^{cl}(\cdot)$, as well as the dimensional mask $\mathcal{M}$ are trained in a meta-learning manner consisting of two steps. In the first regular training step, we train $f_\theta(\cdot)$, $g^{drr}(\cdot)$ and $g_\vartheta^{cl}(\cdot)$ by jointly minimizing the redundancy-reduction and contrastive losses, which is formalized by

$$\mathcal{L}_{regular} = \mathcal{L}_{drr} + \alpha \cdot \mathcal{L}_{contrast} \tag{5}$$

where $\alpha$ is a coefficient that controls the balance between $\mathcal{L}_{drr}$ and $\mathcal{L}_{contrast}$, and we evaluate the influence of this hyper-parameter in Section 6.

In the second meta-learning-based step, we update $\mathcal{M}$ by using the second-derivative technique [44] to solve a bi-level optimization problem. We encourage $\mathcal{M}$ to *mask* (reduce the gradient weights of) the specific dimensions containing task-irrelevant information, which are regarded as dimensional confounders, so that the encoders can better explore the discriminative information in training. To this end, $\mathcal{M}$ can be updated by computing its gradients with respect to the performance of $f_\theta(\cdot)$ and $g_\vartheta^{cl}(\cdot)$. The corresponding performance is measured by using the gradients of $f_\theta(\cdot)$ and $g_\vartheta^{cl}(\cdot)$ during the back-propagation of contrastive loss. Formally, we update the dimensional mask $\mathcal{M}$ by

$$\arg\min_{\mathcal{M}} \mathcal{L}_{contrast} \left( g_{\vartheta_{trial}}^{cl} \left( f_{\theta_{trial}} \left( \mathcal{X}' \right) \otimes \mathcal{M} \right) \right) \tag{6}$$

**Algorithm 1** MetaMask

---

**Input:** Multi-view dataset $\mathcal{X}$, minibatch size $N$, and a hyper-parameter $\alpha$.

**Initialize** The neural network parameters: $\theta$ for $f_\theta(\cdot)$, $\vartheta$ for $g_\vartheta^{cl}(\cdot)$, $\vartheta^{drr}$ for $g^{drr}(\cdot)$, and $\mathcal{M} = \{\omega_k | k \in [\![1, D]\!]\}$. The learning rates: $\ell_\theta$ and $\ell_\vartheta$, etc.

**repeat**

    **for** $t$-th training iteration **do**

        Iteratively sample a minibatch $\mathcal{X}'$ with $N$ examples from $\mathcal{X}$.

        $\# \ regular \ training \ step, \ fix \ \mathcal{M}$

        $\arg \min_{\theta, \vartheta, \vartheta^{drr}} \mathcal{L}_{drr} \left( g_{\vartheta^{drr}}^{drr} \left( f_\theta \left( \mathcal{X}' \right) \otimes \mathcal{M} \right) \right) + \alpha \cdot \mathcal{L}_{contrast} \left( g_\vartheta^{cl} \left( f_\theta \left( \mathcal{X}' \right) \otimes \mathcal{M} \right) \right)$

        $\# \ compute \ trial \ weights \ and \ retain \ computational \ graph \ , \ fix \ \theta \ and \ \vartheta$

        $\theta_{trial} = \theta - \ell_\theta \nabla_\theta \mathcal{L}_{contrast} \left( g_\vartheta^{cl} \left( f_\theta \left( \mathcal{X}' \right) \otimes \mathcal{M} \right) \right)$

        $\vartheta_{trial} = \vartheta - \ell_\vartheta \nabla_\vartheta \mathcal{L}_{contrast} \left( g_\vartheta^{cl} \left( f_\theta \left( \mathcal{X}' \right) \otimes \mathcal{M} \right) \right)$

        $\# \ meta \ training \ step \ using \ second \ derivative$

        $\arg \min_{\mathcal{M}} \mathcal{L}_{contrast} \left( g_{\vartheta_{trial}}^{cl} \left( f_{\theta_{trial}} \left( \mathcal{X}' \right) \otimes \mathcal{M} \right) \right)$

    **end for**

**until** $\theta$, $\vartheta$, $\vartheta^{drr}$, and $\mathcal{M}$ converge.

---

where $\mathcal{X}'$ represents a minibatch sampled from the training dataset $\mathcal{X}$, $\vartheta_{trial}$ and $\theta_{trial}$ represent the *trial* weights of the encoders and projection heads after one gradient update using the contrastive loss defined in Equation 1, respectively. We formulate the updating of such trial weights as follows:

$$\theta_{trial} = \theta - \ell_\theta \nabla_\theta \mathcal{L}_{contrast} \left( g_\vartheta^{cl} \left( f_\theta \left( \mathcal{X}' \right) \otimes \mathcal{M} \right) \right), \vartheta_{trial} = \vartheta - \ell_\vartheta \nabla_\vartheta \mathcal{L}_{contrast} \left( g_\vartheta^{cl} \left( f_\theta \left( \mathcal{X}' \right) \otimes \mathcal{M} \right) \right) \tag{7}$$

where $\ell_\theta$ and $\ell_\vartheta$ are learning rates. The intuition behind such a behavior is that we perform a derivative over the derivative (Hessian matrix) of the combination $\{\theta, \vartheta\}$ to update $\mathcal{M}$, i.e., the second-derivative trick. Specifically, we compute the derivative with respect to $\mathcal{M}$ by using a retained computational graph of $\{\theta, \vartheta\}$ and then update $\mathcal{M}$ by back-propagating the derivative as Equation 6.

We train $\mathcal{M}$ only based on the performance of contrastive learning for two reasons: 1) the dimensional redundancy reduction is a regularization term with strong constraints and generally stable convergence, and it is independent of downstream tasks; 2) such a meta-learning-based training approach that promotes contrastive learning performance empowers our method to constrain the upper and lower bounds of the cross-entropy loss on downstream tasks, which ensures that $\mathcal{M}$ can partially mask the gradient contributions of dimensions containing task-irrelevant information and further promote the encoder to focus on learning task-relevant information, which is demonstrated in Section 5.

The two steps for updating $f_\theta(\cdot)$, $g^{drr}(\cdot)$, and $g_\vartheta^{cl}(\cdot)$ and updating $\mathcal{M}$ are iteratively repeated until convergence. The training pipeline is detailed by Algorithm 1.

## 5 Theoretical Analyses

Current contrastive methods follow the fundamental assumption of multi-view learning (Assumption 1 in [39], Assumption 1 in [40], and Assumption 4.1 in [45]), i.e., views (or distortions) of a same sample holds the invariant discriminative label-relevant information. Theorem 4.2 in [45] further proposes the guarantees for general encoders in the learning paradigm of regular contrastive learning, which proves that the contrastive loss in the self-supervised learning stage can constrain the upper and lower bounds of the cross-entropy loss in the supervised learning stage for downstream tasks. To elaborate the behavior of our MetaMask, we extend this Theorem to build a connection between the proposed masked representation and downstream performance by

**Theorem 5.1.** *(Connecting Masked Representations to Downstream Cross-Entropy Loss). Under the general assumption of self-supervised multi-view learning, when $\mathcal{M}$ is optimal, for any coupled $\boldsymbol{h}, \tilde{\boldsymbol{h}} \in \mathbb{R}$, $\mathcal{L}_{crossentropy}\left( \boldsymbol{h} \right)$ for downstream classification can be bounded by $\mathcal{L}_{contrast}\left( g_\vartheta^{cl}\left( \tilde{\boldsymbol{h}} \right) \right)$*

Table 1: Comparison of different methods on classification accuracy (top 1). We follow the experimental setting of [8] and adopt *conv* and *fc* as the backbones in the experiments.

| Model | IN-200 [46] | | STL-10 [47] | | CIFAR-10 [46] | | CIFAR-100 [46] | |
|---|---|---|---|---|---|---|---|---|
| | conv | fc | conv | fc | conv | fc | conv | fc |
| Barlow Twins [2] | 39.81 | 40.34 | 80.97 | 81.43 | 76.63 | 78.49 | 52.80 | 52.95 |
| + MetaMask$_{+4.95}$ | 42.91$_{+3.10}$ | 42.06$_{+1.72}$ | 84.96$_{+3.99}$ | 86.13$_{+4.70}$ | 85.24$_{+8.61}$ | 86.90$_{+8.41}$ | 58.02$_{+5.22}$ | 56.78$_{+3.83}$ |
| SwAV [6] | 39.67 | 39.02 | 74.65 | 75.30 | 72.19 | 72.36 | 46.58 | 46.75 |
| + MetaMask$_{+5.00}$ | 40.28$_{+0.61}$ | 40.05$_{+1.03}$ | 81.34$_{+6.69}$ | 82.19$_{+6.89}$ | 80.37$_{+8.18}$ | 77.22$_{+4.86}$ | 52.07$_{+5.49}$ | 52.98$_{+6.23}$ |
| SimCLR [1] | 36.24 | 39.83 | 75.57 | 77.15 | 80.58 | 80.07 | 50.03 | 49.82 |
| + MetaMask$_{+3.60}$ | 40.56$_{+4.32}$ | 40.82$_{+0.99}$ | 81.37$_{+5.80}$ | 82.90$_{+5.75}$ | 82.85$_{+2.27}$ | 81.64$_{+1.57}$ | 53.70$_{+3.67}$ | 54.24$_{+4.42}$ |
| CMC [8] | 41.58 | 40.11 | 83.03 | 85.06 | 81.31 | 83.28 | 58.13 | 56.72 |
| + MetaMask$_{+1.73}$ | 42.91$_{+1.33}$ | 42.06$_{+1.95}$ | 84.96$_{+1.93}$ | 86.13$_{+1.07}$ | 85.24$_{+3.93}$ | 86.90$_{+3.62}$ | 58.02$_{-0.09}$ | 56.78$_{+0.06}$ |
| BYOL [14] | 41.59 | 41.90 | 81.73 | 81.57 | 77.18 | 80.01 | 53.64 | 53.78 |
| + MetaMask$_{+1.68}$ | 42.72$_{+1.13}$ | 43.29$_{+1.39}$ | 83.01$_{+1.28}$ | 83.44$_{+1.87}$ | 79.26$_{+2.08}$ | 82.73$_{+2.72}$ | 55.68$_{+2.04}$ | 54.72$_{+0.94}$ |
| SimSiam [16] | 41.03 | 41.27 | 80.91 | 81.88 | 78.14 | 81.13 | 52.55 | 53.52 |
| + MetaMask$_{+1.44}$ | 41.82$_{+0.79}$ | 42.67$_{+1.40}$ | 82.35$_{+1.44}$ | 83.96$_{+2.08}$ | 80.03$_{+1.89}$ | 81.39$_{+0.26}$ | 54.70$_{+2.15}$ | 55.02$_{+1.50}$ |
| DCL [10] | 38.79 | 40.26 | 77.09 | 78.39 | 80.89 | 80.93 | 51.38 | 51.09 |
| + MetaMask$_{+1.83}$ | 40.28$_{+1.49}$ | 41.69$_{+1.43}$ | 80.94$_{+3.85}$ | 81.06$_{+2.67}$ | 81.33$_{+0.44}$ | 81.34$_{+0.41}$ | 53.05$_{+1.67}$ | 53.74$_{+2.65}$ |
| HCL [11] | 40.05 | 41.23 | 79.86 | 80.20 | 82.13 | 82.76 | 52.69 | 53.13 |
| + MetaMask$_{+1.14}$ | 40.43$_{+0.38}$ | 41.01$_{-0.22}$ | 81.52$_{+1.66}$ | 82.09$_{+1.89}$ | 83.65$_{+1.52}$ | 83.95$_{+1.19}$ | 54.03$_{+1.34}$ | 54.26$_{+1.13}$ |

*in self-supervised learning:*

$$\mathcal{L}_{contrast}\left(g_\vartheta^{cl}\left(\tilde{\boldsymbol{h}}\right)\right) - \sqrt{\Phi\left(g_\vartheta^{cl}\left(\tilde{\boldsymbol{h}}\right)\Big|\boldsymbol{y}\right)} - \frac{1}{2}\cdot\sum_{k\in[\![1,D]\!]}\sqrt{\Phi\left(\lfloor g_\vartheta^{cl}\left(\tilde{\boldsymbol{h}}\right)\rceil^k\Big|\boldsymbol{y}\right)} - \mathcal{O}\left(M^{-\frac{1}{2}}\right)$$

$$\leq \mathcal{L}_{crossentropy}\left(\boldsymbol{h}\right) + \log\left(\frac{M}{D}\right) \leq \mathcal{L}_{contrast}\left(g_\vartheta^{cl}\left(\tilde{\boldsymbol{h}}\right)\right) + \sqrt{\Phi\left(g_\vartheta^{cl}\left(\tilde{\boldsymbol{h}}\right)\Big|\boldsymbol{y}\right)} + \mathcal{O}\left(M^{-\frac{1}{2}}\right)$$

$$(8)$$

*where $M$ denotes the quantity of negative samples, $D$ denotes the dimensionality of the representation, $\boldsymbol{h}$ denotes the unmasked representation, $\tilde{\boldsymbol{h}}$ denotes the masked representation obtained by Equation 4, $\boldsymbol{y}$ denotes the target label, $\Phi\left(\cdot|\boldsymbol{y}\right) = \mathbb{E}_{\mathcal{P}(y)}\left[\mathbb{E}_{\mathcal{P}(\cdot|y)}\delta\left(\cdot,\mathbb{E}_{\mathcal{P}(\cdot|y)}\right)\right]$ represents the conditional variance function where $\delta\left(\cdot,\cdot\right)$ is a discriminating function to measure the difference between the terms, e.g., $\delta\left(\cdot,\cdot\right) = \|[\cdot]-[\cdot]\|^2$ in low-dimensional embedding space or $\delta\left(\cdot,\cdot\right) = -\log\left(\cos\left(\cdot,\cdot\right)\right) = -\log\frac{[\cdot]\times[\cdot]}{\|\cdot\|\times\|\cdot\|}$ in high-dimensional embedding space, $\lfloor\cdot\rceil^k$ is a function acquiring $k$-th dimension vector, $\log\left(\frac{M}{D}\right)$ is a constant, and $\mathcal{O}\left(M^{-\frac{1}{2}}\right)$ is the approximation error's order.*

Theorem 5.1 states that reducing the risk of contrastive learning can improve the performance on downstream tasks, which further supports our intuition to make the model focus on the acquisition of discriminative information by learning a dimensional mask with the objective of improving the performance of masked representations on contrastive learning is theoretically sound. Refer to Appendix A.1 for the elaboration of MetaMask's behavior to mask the gradient contribution of specific dimensions containing confounder information. Hence, we derive

**Theorem 5.2.** *(Guarantees for Reduced Conditional Variance of Masked Representations). When $\mathcal{M}$ is optimal, for any coupled $\boldsymbol{h}, \tilde{\boldsymbol{h}} \in \mathbb{R}$, given label $\boldsymbol{y}$, the conditional variance of $\tilde{\boldsymbol{h}}$ is reduced:*

$$\Phi\left(g_\vartheta^{cl}\left(\tilde{\boldsymbol{h}}\right)\Big|\boldsymbol{y}\right) \leq \Phi\left(g_\vartheta^{cl}\left(\boldsymbol{h}\right)\Big|\boldsymbol{y}\right), \; yet \;\; \Phi\left(\lfloor g_\vartheta^{cl}\left(\tilde{\boldsymbol{h}}\right)\rceil^k\Big|\boldsymbol{y}\right) \cong \Phi\left(\lfloor g_\vartheta^{cl}\left(\boldsymbol{h}\right)\rceil^k\Big|\boldsymbol{y}\right). \qquad (9)$$

Theorem 5.2 states that given the label $y$, the masked representation $\tilde{\boldsymbol{h}}$ has smaller conditional variance than $\boldsymbol{h}$ in contrastive learning. We bring Theorem 5.2 into Theorem 5.1 to derive a conclusion: our approach can better bound the downstream classification risk, i.e., the upper and lower bounds of supervised cross-entropy loss obtained by MetaMask are tighter than typical contrastive learning methods. Please refer to Appendix A.2 for the corresponding proofs.

# 6   Experiments

## 6.1   Benchmarking MetaMask with Various Backbones

**Implementations.** For the comparisons demonstrated in Table 1, we uniformly set the batch size as 64, and we adopt a network with the 5 convolutional layers in AlexNet [48] as *conv* and a network with 2 additional fully connected layers as *fc*. We straightforwardly adopt the same data augmentations in [8] and [49], and the memory bank [50] is adopted to facilitate calculations by retrieving 4,096 past negative features during training. Following the experimental principle of linear probing, we remove the projection head and re-train an MLP as the classification head for the learned representation.

For the experiments in Table 2, the batch size is valued by 512, and ResNet-18 [51] is used as the backbone encoder. We adopt the data augmentation and other experimental settings following [2]. Note that the KNN classifier is adopted for evaluation.

**Discussion of the comparisons using conv and fc encoders.** Table 1 shows the comparisons on four benchmark datasets, and the corresponding positive or negative margins are reported. We observe that MetaMask beats the best prior methods by a significant margin on most downstream tasks, which demonstrates that MetaMask can efficiently improve the performance of the learned representation when the self-supervision is insufficient, i.e., the limited batch size and weak encoder.

**Discussion of the comparisons using the ResNet encoder.** We further perform classification comparisons using a strong encoder, i.e., ResNet. Table 2 shows that MetaMask can consistently improve the state-of-the-art methods, which indicates that the proposed method has strong adaptability to different encoders. The experimental evidence proves our assumption of the dimensional confounder. Thus, given a fixed dimensionality of the projection head, MetaMask can indeed improve the discriminativeness of the learned representation by *masking* the dimensional confounder.

## 6.2   Validation of the Robustness of MetaMask against Dimensional Confounder

We conduct experiments to prove that MetaMask is able to mitigate the negative impact of the dimensional confounder on the discriminativeness of the learned representations, and the results are shown in Figure 4. The intriguing outcome demonstrates two merits of the proposed method: 1) even if the dimensionality of the projection head changes, MetaMask is consistently robust to the dimensional confounder, i.e., the model maintains relatively consistent performance; 2) MetaMask can alleviate the negative impact of *curse of dimensionality* [52, 53], e.g., given a fixed number of examples, as the dimensionality of the representation increases during training, the performance of the model on downstream tasks instead degenerates. Concretely, Figure 2 demonstrates that regardless of whether representation learning suffers from the curse of dimensionality, dimensionality confounding factors are widely present in learned representations, and hence, the defined dimensional confounder is a more generalized issue of the curse of dimensionality. Refer Appendix A.3 for the further discussion. Moreover, we observe a surprising result that MetaMask* achieves better performance. We provide an understandable explanation: the essence of the curse of dimensionality is that a network with a large number of parameters is prone to over-fitting to the training data. Therefore, compared with the annealing strategy, a fixed and relatively large learning rate can prevent the model from falling into a local optimum (over-fitting) during optimization. Our further experiments show that MetaMask* collapses abruptly at larger dimensional settings, e.g., 16384, which demonstrates that the trick of using a fixed learning rate can only unsteadily improve the performance of MetaMask, but this behavior is *not* essential. Figure 4 shows that under the different experimental settings, vanilla MetaMask consistently counteracts the negative effects of the dimensional confounder.

# 7   Conclusion and Future Discussion

From the dimensional perspective, we first clarify that the dimensional redundancy and confounder issues are the essence of the current self-supervised learning paradigm's defects. Multiple motivating experiments are conducted to support our viewpoint, and the intriguing results are opposite to the traditional conclusions. To jointly tackle the dimensional redundancy and confounder issues, we propose a heuristic learning approach, called MetaMask, which leverages a second-derivative technique to learn a dimensional mask in order to adjust the gradient weights of different dimensions during training. The proposed theoretical analyses demonstrate that MetaMask obtains relatively

Table 2: Comparison of different methods on classification accuracy (top 1). We adopt *ResNet-18* as the backbones.

| Model | CIFAR-10 | CIFAR-100 | STL-10 | IN-200 |
|---|---|---|---|---|
| Barlow Twins [2] | 85.72 | 60.31 | 73.62 | 45.10 |
| + MetaMask$_{+1.25}$ | 87.53$_{+1.81}$ | 61.42$_{+1.11}$ | 73.97$_{+0.35}$ | 46.81$_{+1.71}$ |
| SimCLR [1] | 81.73 | 58.27 | 70.09 | 44.46 |
| + MetaMask$_{+3.32}$ | 86.01$_{+4.28}$ | 61.03$_{+2.76}$ | 74.90$_{+4.81}$ | 45.87$_{+1.41}$ |
| BYOL [14] | 88.05 | 60.94 | 72.04 | 46.72 |
| + MetaMask$_{+0.28}$ | 87.53$_{-0.52}$ | 61.42$_{+0.48}$ | 73.97$_{+1.93}$ | 46.81$_{+0.11}$ |
| DCL [10] | 85.61 | 59.29 | 71.18 | 45.77 |
| + MetaMask$_{+1.65}$ | 86.47$_{+0.86}$ | 60.82$_{+1.53}$ | 74.79$_{+3.61}$ | 46.35$_{+0.58}$ |
| HCL [11] | 85.27 | 61.21 | 71.92 | 46.90 |
| + MetaMask$_{+0.95}$ | 85.97$_{+0.70}$ | 61.63$_{+0.42}$ | 74.03$_{+2.11}$ | 46.06$_{+0.55}$ |
| NNCLR [54] | 81.44 | 61.64 | 71.89 | 47.10 |
| + MetaMask$_{+1.63}$ | 85.78$_{+4.34}$ | 61.89$_{+0.25}$ | 74.02$_{+2.13}$ | 46.90$_{-0.20}$ |

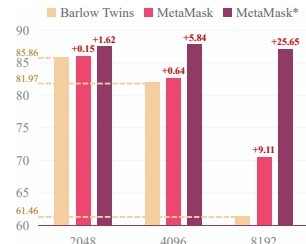

Figure 4: Comparisons of varying the dimensionality of the projection head on CIFAR-10 by using ResNet-18. ∗ denotes MetaMask using a trick of fixed learning rate instead of the cosine annealing strategy.

tighter risk bounds for downstream classification compared to conventional contrastive methods. From the experiments, we observe a consistent improvement in the performance of our method over state-of-the-art methods on benchmark datasets.

One of our critical contributions is proposing a heuristic learning paradigm for self-supervised learning: regarding the dimensional redundancy reduction as a regularization term and further exploring to mask dimensions containing the confounder. In the current methodology, we train the learnable dimensional mask with the objective of improving the performance of masked representations on a specific *self-supervised task*, i.e., contrastive learning, and provide the theoretical analysis to prove that such a learning approach can indeed prompt the model focus on acquiring discriminative information. However, contrastive learning is immutable for the *self-supervised task* in MetaMask. It is commonly acknowledged that masked image modeling (MIM) methods [55, 56, 57] achieve impressive performance in self-supervised learning. Although the theoretical contribution bridging the gap between MIM loss and cross-entropy loss on downstream tasks has not yet emerged, we still deem combining MIM approaches with our proposed paradigm is attractive to the community.

## Acknowledgements

The authors would like to thank the anonymous reviewers for their valuable comments. This work is supported in part by National Natural Science Foundation of China No. 61976206 and No. 61832017, Key Special Project for Introduced Talents Team of Southern Marine Science and Engineering Guangdong Laboratory (Guangzhou) No. GML2019ZD0603, National Key Research and Development Program of China No. 2019YFB1405100, Foshan HKUST Projects (FSUST21-FYTRI01A, FSUST21-FYTRI02A), Beijing Outstanding Young Scientist Program NO. BJJWZYJH012019100020098, Beijing Academy of Artificial Intelligence (BAAI), the Fundamental Research Funds for the Central Universities, the Research Funds of Renmin University of China 21XNLG05, and Public Computing Cloud, Renmin University of China. This work is also supported in part by Intelligent Social Governance Platform, Major Innovation & Planning Interdisciplinary Platform for the "Double-First Class" Initiative, Renmin University of China, and Public Policy and Decision-making Research Lab of Renmin University of China.

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
