# A  Appendix

## A.1  Elaborating MetaMask's Behavior of Masking Gradient Contribution

For better understanding the intuition behind the behavior of the proposed MetaMask, we elaborate the procedure of our method masking gradient contributions of dimensions. We restate Equation 1 of contrastive loss by converting the original projected features $\boldsymbol{z}$ into the masked feature $\tilde{\boldsymbol{z}}$:

$$
\begin{aligned}
\mathcal{L}_{maskedcl} &= -\sum\sum \log \frac{\exp\left[d\left(\tilde{\boldsymbol{z}}, \tilde{\boldsymbol{z}}^{+}\right)/\tau\right]}{\sum\sum \mathbb{1}\cdot\exp\left[d\left(\tilde{\boldsymbol{z}}, \tilde{\boldsymbol{z}}'\right)/\tau\right]} \\
&= -\sum\sum \log \frac{\exp\left[d\left(g_{\vartheta}^{cl}\left(\tilde{\boldsymbol{h}}\right), g_{\vartheta}^{cl}\left(\tilde{\boldsymbol{h}}^{+}\right)\right)/\tau\right]}{\sum\sum \mathbb{1}\cdot\exp\left[d\left(g_{\vartheta}^{cl}\left(\tilde{\boldsymbol{h}}\right), g_{\vartheta}^{cl}\left(\tilde{\boldsymbol{h}}'\right)\right)/\tau\right]} \\
&= -\sum\sum \log \frac{\exp\left[d\left(g_{\vartheta}^{cl}\left(\boldsymbol{h}\otimes\mathcal{M}\right), g_{\vartheta}^{cl}\left(\boldsymbol{h}^{+}\otimes\mathcal{M}\right)\right)/\tau\right]}{\sum\sum \mathbb{1}\cdot\exp\left[d\left(g_{\vartheta}^{cl}\left(\boldsymbol{h}\otimes\mathcal{M}\right), g_{\vartheta}^{cl}\left(\boldsymbol{h}'\otimes\mathcal{M}\right)\right)/\tau\right]},
\end{aligned}
\tag{10}
$$

which is derived by considering Equation 4. To simplify the equation, we hold

$$
\mathcal{L}_{maskedcl} = \sum_{i\in[\![1,N]\!]} \sum_{\substack{j\in[\![1,M-1]\!] \\ j^{+}\in[\![j+1,M]\!]}} \mathcal{L}_{i,j,j^{+}}.
\tag{11}
$$

Therefore, the indicator function $\mathbb{1}_{[i\neq i'\vee j\neq j']}$ can be substituted as follows:

$$
\mathcal{L}_{i,j,j^{+}} = \log \frac{-\exp\left[d\left(g\left(\boldsymbol{h}\otimes\mathcal{M}\right), g\left(\boldsymbol{h}^{+}\otimes\mathcal{M}\right)\right)/\tau\right]}{\exp\left[d\left(g\left(\boldsymbol{h}\otimes\mathcal{M}\right), g\left(\boldsymbol{h}^{+}\otimes\mathcal{M}\right)\right)/\tau\right] + \sum_{\boldsymbol{h}^{-}}\exp\left[d\left(g\left(\boldsymbol{h}\otimes\mathcal{M}\right), g\left(\boldsymbol{h}^{-}\otimes\mathcal{M}\right)\right)/\tau\right]},
\tag{12}
$$

where $g\left(\cdot\right)$ denotes $g_{\vartheta}^{cl}\left(\cdot\right)$, and $\boldsymbol{h}^{-}$ is neither equal to $\boldsymbol{h}$ nor equal to $\boldsymbol{h}^{+}$.

**Proposition A.1.** *The mask $\mathcal{M}$ exists on each partial differential equation of the gradient of $\mathcal{L}_{i,j,j^{+}}$:*

$$
\begin{aligned}
\nabla_{\boldsymbol{h}}\mathcal{L}_{i,j,j^{+}} =& \frac{\sum_{\boldsymbol{h}^{-}}\exp\left[d\left(g\left(\boldsymbol{h}\otimes\mathcal{M}\right), g\left(\boldsymbol{h}^{-}\otimes\mathcal{M}\right)\right)/\tau\right] - \exp\left[d\left(g\left(\boldsymbol{h}\otimes\mathcal{M}\right), g\left(\boldsymbol{h}^{+}\otimes\mathcal{M}\right)\right)/\tau\right]}{\sum_{\boldsymbol{h}^{-}}\exp\left[d\left(g\left(\boldsymbol{h}\otimes\mathcal{M}\right), g\left(\boldsymbol{h}^{-}\otimes\mathcal{M}\right)\right)/\tau\right]} \\
&\times \left[\sum_{\boldsymbol{h}^{-'}}\frac{\exp\left[d\left(g\left(\boldsymbol{h}\otimes\mathcal{M}\right), g\left(\boldsymbol{h}^{-'}\otimes\mathcal{M}\right)\right)/\tau\right]}{\sum_{\boldsymbol{h}^{-}}\exp\left[d\left(g\left(\boldsymbol{h}\otimes\mathcal{M}\right), g\left(\boldsymbol{h}^{-}\otimes\mathcal{M}\right)\right)/\tau\right]} \times g\left(\boldsymbol{h}^{-'}\otimes\mathcal{M}\right) - g\left(\boldsymbol{h}^{+}\otimes\mathcal{M}\right)\right] \\
&\times \prod_{k\in[\![1,N^{g}]\!]} \boldsymbol{w}_{k} \times \nabla_{\boldsymbol{h}}\boldsymbol{h}\otimes\mathcal{M}
\end{aligned}
\tag{13}
$$

$$
\begin{aligned}
\nabla_{\boldsymbol{h}^{+}}\mathcal{L}_{i,j,j^{+}} =& \frac{\exp\left[d\left(g\left(\boldsymbol{h}\otimes\mathcal{M}\right), g\left(\boldsymbol{h}^{+}\otimes\mathcal{M}\right)\right)/\tau\right] - \sum_{\boldsymbol{h}^{-}}\exp\left[d\left(g\left(\boldsymbol{h}\otimes\mathcal{M}\right), g\left(\boldsymbol{h}^{-}\otimes\mathcal{M}\right)\right)/\tau\right]}{\sum_{\boldsymbol{h}^{-}}\exp\left[d\left(g\left(\boldsymbol{h}\otimes\mathcal{M}\right), g\left(\boldsymbol{h}^{-}\otimes\mathcal{M}\right)\right)/\tau\right]} \\
&\times g\left(\boldsymbol{h}\otimes\mathcal{M}\right) \times \prod_{k\in[\![1,N^{g}]\!]} \boldsymbol{w}_{k} \times \nabla_{\boldsymbol{h}^{+}}\boldsymbol{h}^{+}\otimes\mathcal{M}
\end{aligned}
\tag{14}
$$

$$
\begin{aligned}
\nabla_{\boldsymbol{h}^{-}}\mathcal{L}_{i,j,j^{+}} =& \frac{\sum_{\boldsymbol{h}^{-}}\exp\left[d\left(g\left(\boldsymbol{h}\otimes\mathcal{M}\right), g\left(\boldsymbol{h}^{-}\otimes\mathcal{M}\right)\right)/\tau\right] - \exp\left[d\left(g\left(\boldsymbol{h}\otimes\mathcal{M}\right), g\left(\boldsymbol{h}^{+}\otimes\mathcal{M}\right)\right)/\tau\right]}{\sum_{\boldsymbol{h}^{-}}\exp\left[d\left(g\left(\boldsymbol{h}\otimes\mathcal{M}\right), g\left(\boldsymbol{h}^{-}\otimes\mathcal{M}\right)\right)/\tau\right]} \\
&\times \frac{\exp\left[d\left(g\left(\boldsymbol{h}\otimes\mathcal{M}\right), g\left(\boldsymbol{h}^{-}\otimes\mathcal{M}\right)\right)/\tau\right]}{\sum_{\boldsymbol{h}^{-'}}\exp\left[d\left(g\left(\boldsymbol{h}\otimes\mathcal{M}\right), g\left(\boldsymbol{h}^{-'}\otimes\mathcal{M}\right)\right)/\tau\right]} \times g\left(\boldsymbol{h}\otimes\mathcal{M}\right) \times \prod_{k\in[\![1,N^{g}]\!]} \boldsymbol{w}_{k} \\
&\times \nabla_{\boldsymbol{h}^{-}}\boldsymbol{h}^{-}\otimes\mathcal{M}
\end{aligned}
\tag{15}
$$

*where $\left\{\boldsymbol{w}_k \big| k \in \llbracket 1, N^g \rrbracket\right\}$ denotes the weights of the projection head $g\left(\cdot\right)$, $N^g$ denotes the number of corresponding projection layers, and $\boldsymbol{h}^{-'}$ is neither equal to $\boldsymbol{h}$ nor equal to $\boldsymbol{h}^+$ (note that $\boldsymbol{h}^-$ and $\boldsymbol{h}^{-'}$ are two independent variables). $\mathcal{M}$ is the dimensional mask.*

Proposition A.1 states that the dimensional mask $\mathcal{M}$ consistently exists on each partial differential equation of the gradient of the contrastive loss $\mathcal{L}_{i,j,j^+}$. We further observe that there exists 7 types of irreducible terms containing $\mathcal{M}$: $g\left(\boldsymbol{h} \otimes \mathcal{M}\right)$, $g\left(\boldsymbol{h}^+ \otimes \mathcal{M}\right)$, $g\left(\boldsymbol{h}^- \otimes \mathcal{M}\right)$, $g\left(\boldsymbol{h}^{-'} \otimes \mathcal{M}\right)$, $\nabla_{\boldsymbol{h}}\boldsymbol{h} \otimes \mathcal{M}$, $\nabla_{\boldsymbol{h}^+}\boldsymbol{h} \otimes \mathcal{M}$, $\nabla_{\boldsymbol{h}^-}\boldsymbol{h} \otimes \mathcal{M}$, where the first 4 types are in the regular computing manner so that $\mathcal{M}$ can directly change the values of different dimensions by assigning dimension-specific weights, and the last 3 types are the first-order derivatives. Note that the scalar derivation, e.g., the loss, of a matrix can be thought of as derivation of each element of the matrix, and such a derivation is derived and then putting the element-wise results in the order of the matrix to get a gradient matrix of the same dimension.

Following such a law, we utilize Chain Rules to derive the derivative of the loss $\mathcal{L}_{i,j,j^+}$ with respect to each element in the batch matrix containing $\boldsymbol{h}$, $\boldsymbol{h}^+$, $\boldsymbol{h}^-$ and further obtain the derivative matrix:

$$\frac{\partial \mathcal{L}_{i,j,j^+}}{\partial \mathcal{X}'_{p,q}} = \frac{\partial \mathcal{L}_{i,j,j^+}}{\partial \mathcal{B}_{p,q}}\frac{\partial \mathcal{B}_{p,q}}{\partial \mathcal{X}'_{p,q}} = \frac{\partial \mathcal{L}_{i,j,j^+}}{\partial \left[\left(\mathcal{B} \otimes \hat{\mathcal{M}}\right)_{p,q}\right]}\frac{\partial \left[\left(\mathcal{B} \otimes \hat{\mathcal{M}}\right)_{p,q}\right]}{\partial \mathcal{B}_{p,q}}\frac{\partial \mathcal{B}_{p,q}}{\partial \mathcal{X}'_{p,q}}$$

$$= \begin{bmatrix} \boldsymbol{\omega_1}\frac{\partial \lfloor f_\theta(\boldsymbol{x})\rceil^1}{\partial \boldsymbol{x}} & \boldsymbol{\omega_2}\frac{\partial \lfloor f_\theta(\boldsymbol{x})\rceil^2}{\partial \boldsymbol{x}} & \cdots & \boldsymbol{\omega_{D-1}}\frac{\partial \lfloor f_\theta(\boldsymbol{x})\rceil^{D-1}}{\partial \boldsymbol{x}} & \boldsymbol{\omega_D}\frac{\partial \lfloor f_\theta(\boldsymbol{x})\rceil^D}{\partial \boldsymbol{x}} \\ \boldsymbol{\omega_1}\frac{\partial \lfloor f_\theta(\boldsymbol{x}^+)\rceil^1}{\partial \boldsymbol{x}^+} & \boldsymbol{\omega_2}\frac{\partial \lfloor f_\theta(\boldsymbol{x}^+)\rceil^2}{\partial \boldsymbol{x}^+} & \cdots & \boldsymbol{\omega_{D-1}}\frac{\partial \lfloor f_\theta(\boldsymbol{x}^+)\rceil^{D-1}}{\partial \boldsymbol{x}^+} & \boldsymbol{\omega_D}\frac{\partial \lfloor f_\theta(\boldsymbol{x}^+)\rceil^D}{\partial \boldsymbol{x}^+} \\ \boldsymbol{\omega_1}\frac{\partial \lfloor f_\theta(\boldsymbol{x}_1^-)\rceil^1}{\partial \boldsymbol{x}_1^-} & \boldsymbol{\omega_2}\frac{\partial \lfloor f_\theta(\boldsymbol{x}_1^-)\rceil^2}{\partial \boldsymbol{x}_1^-} & \cdots & \boldsymbol{\omega_{D-1}}\frac{\partial \lfloor f_\theta(\boldsymbol{x}_1^-)\rceil^{D-1}}{\partial \boldsymbol{x}_1^-} & \boldsymbol{\omega_D}\frac{\partial \lfloor f_\theta(\boldsymbol{x}_1^-)\rceil^D}{\partial \boldsymbol{x}_1^-} \\ \boldsymbol{\omega_1}\frac{\partial \lfloor f_\theta(\boldsymbol{x}_2^-)\rceil^1}{\partial \boldsymbol{x}_2^-} & \boldsymbol{\omega_2}\frac{\partial \lfloor f_\theta(\boldsymbol{x}_2^-)\rceil^2}{\partial \boldsymbol{x}_2^-} & \cdots & \boldsymbol{\omega_{D-1}}\frac{\partial \lfloor f_\theta(\boldsymbol{x}_2^-)\rceil^{D-1}}{\partial \boldsymbol{x}_2^-} & \boldsymbol{\omega_D}\frac{\partial \lfloor f_\theta(\boldsymbol{x}_2^-)\rceil^D}{\partial \boldsymbol{x}_2^-} \\ \cdots & \cdots & \cdots & \cdots & \cdots \\ \boldsymbol{\omega_1}\frac{\partial \lfloor f_\theta(\boldsymbol{x}_{N-3}^-)\rceil^1}{\partial \boldsymbol{x}_{N-3}^-} & \boldsymbol{\omega_2}\frac{\partial \lfloor f_\theta(\boldsymbol{x}_{N-3}^-)\rceil^2}{\partial \boldsymbol{x}_{N-3}^-} & \cdots & \boldsymbol{\omega_{D-1}}\frac{\partial \lfloor f_\theta(\boldsymbol{x}_{N-3}^-)\rceil^{D-1}}{\partial \boldsymbol{x}_{N-3}^-} & \boldsymbol{\omega_D}\frac{\partial \lfloor f_\theta(\boldsymbol{x}_{N-3}^-)\rceil^D}{\partial \boldsymbol{x}_{N-3}^-} \\ \boldsymbol{\omega_1}\frac{\partial \lfloor f_\theta(\boldsymbol{x}_{N-2}^-)\rceil^1}{\partial \boldsymbol{x}_{N-2}^-} & \boldsymbol{\omega_2}\frac{\partial \lfloor f_\theta(\boldsymbol{x}_{N-2}^-)\rceil^2}{\partial \boldsymbol{x}_{N-2}^-} & \cdots & \boldsymbol{\omega_{D-1}}\frac{\partial \lfloor f_\theta(\boldsymbol{x}_{N-2}^-)\rceil^{D-1}}{\partial \boldsymbol{x}_{N-2}^-} & \boldsymbol{\omega_D}\frac{\partial \lfloor f_\theta(\boldsymbol{x}_{N-2}^-)\rceil^D}{\partial \boldsymbol{x}_{N-2}^-} \end{bmatrix}$$

$$\tag{16}$$

where $\mathcal{B} = \left[\boldsymbol{h}, \boldsymbol{h}^+, \boldsymbol{h}_1^-, \boldsymbol{h}_2^-, \ldots, \boldsymbol{h}_{N-3}^-, \boldsymbol{h}_{N-2}^-\right]$ is a batch of features with $N$ examples, and $\mathcal{X}'$ is the corresponding input data matrix. $\boldsymbol{h}$, $\boldsymbol{h}^+$, and $\boldsymbol{h}^-$ are feature vectors containing $D$ dimensions. $\hat{\mathcal{M}} = [\mathcal{M}, \mathcal{M}, \ldots, \mathcal{M}, \mathcal{M}]$ is an extended matrix containing $N$ copies of $\mathcal{M}$. Intriguingly, we observe that $\mathcal{M}$ generally assign the dimension-specific weights for each samples, which empowers MetaMask to naturally reduce the gradient effect of specific dimensions containing the confounder by employing the meta-learning paradigm with the objective of improving the performance of masked representations on a typical self-supervised task. After clarifying the dimension-specific weighting mechanism from the perspective of gradient analysis, we further demonstrate the effectiveness of MetaMask's training paradigm as follows.

## A.2 Proofs

In order to prove Theorem 5.2 and the conclusion that the bounds of supervised cross-entropy loss obtained by MetaMask are tighter than typical contrastive learning methods, we provide separated proofs for the corresponding two parts of Theorem 5.2 and the tighter bounds on downstream cross-entropy loss.

### A.2.1 Proof for the Equality Part

To prove

$$\Phi\left(\lfloor g_\vartheta^{cl}\left(\tilde{\boldsymbol{h}}\right)\rceil^k \big| \boldsymbol{y}\right) \cong \Phi\left(\lfloor g_\vartheta^{cl}\left(\boldsymbol{h}\right)\rceil^k \big| \boldsymbol{y}\right), \tag{17}$$

we observe that the only variable that differs on both sides of the Equation 17 is the representation, which is masked on the left hand side of the equation while unmasked on the right hand side of the

equation. Then, we substitute the definition of $\Phi\left(\cdot\right)$ into Equation 17 and derive

$$
\begin{aligned}
&\mathbb{E}_{\mathcal{P}(y)}\left[\mathbb{E}_{\mathcal{P}\left(\lfloor g_{\vartheta}^{cl}(\tilde{\boldsymbol{h}})\rceil^{k}|y\right)}\delta\left(\lfloor g_{\vartheta}^{cl}\left(\tilde{\boldsymbol{h}}\right)\rceil^{k},\mathbb{E}_{\mathcal{P}\left(\lfloor g_{\vartheta}^{cl}(\tilde{\boldsymbol{h}})\rceil^{k}|y\right)}\right)\right]\\
&\cong \mathbb{E}_{\mathcal{P}(y)}\left[\mathbb{E}_{\mathcal{P}\left(\lfloor g_{\vartheta}^{cl}(\boldsymbol{h})\rceil^{k}|y\right)}\delta\left(\lfloor g_{\vartheta}^{cl}\left(\boldsymbol{h}\right)\rceil^{k},\mathbb{E}_{\mathcal{P}\left(\lfloor g_{\vartheta}^{cl}(\boldsymbol{h})\rceil^{k}|y\right)}\right)\right].
\end{aligned}
\tag{18}
$$

In our approach, the learned representation generally exists in high-dimensional embedding space, e.g., 1024, 2048, 8192, etc. Therefore, in high-dimensional embedding space, the left hand side of the equation can be extended to

$$
\begin{aligned}
&\mathbb{E}_{\mathcal{P}(y)}\left[\mathbb{E}_{\mathcal{P}\left(\lfloor g_{\vartheta}^{cl}(\tilde{\boldsymbol{h}})\rceil^{k}|y\right)}\delta\left(\lfloor g_{\vartheta}^{cl}\left(\tilde{\boldsymbol{h}}\right)\rceil^{k},\mathbb{E}_{\mathcal{P}\left(\lfloor g_{\vartheta}^{cl}(\tilde{\boldsymbol{h}})\rceil^{k}|y\right)}\right)\right]\\
&= \mathbb{E}_{\mathcal{P}(y)}\left[\mathbb{E}_{\mathcal{P}\left(\lfloor g_{\vartheta}^{cl}(\tilde{\boldsymbol{h}})\rceil^{k}|y\right)}-\log\left(\cos\left(\lfloor g_{\vartheta}^{cl}\left(\tilde{\boldsymbol{h}}\right)\rceil^{k},\mathbb{E}_{\mathcal{P}\left(\lfloor g_{\vartheta}^{cl}(\tilde{\boldsymbol{h}})\rceil^{k}|y\right)}\right)\right)\right]\\
&= \mathbb{E}_{\mathcal{P}(y)}\left[\mathbb{E}_{\mathcal{P}\left(\lfloor g_{\vartheta}^{cl}(\tilde{\boldsymbol{h}})\rceil^{k}|y\right)}-\log\frac{\lfloor g_{\vartheta}^{cl}\left(\tilde{\boldsymbol{h}}\right)\rceil^{k}\times\mathbb{E}_{\mathcal{P}\left(\lfloor g_{\vartheta}^{cl}(\tilde{\boldsymbol{h}})\rceil^{k}|y\right)}}{||\lfloor g_{\vartheta}^{cl}\left(\tilde{\boldsymbol{h}}\right)\rceil^{k}||\times||\mathbb{E}_{\mathcal{P}\left(\lfloor g_{\vartheta}^{cl}(\tilde{\boldsymbol{h}})\rceil^{k}|y\right)}||}\right]\\
&= \mathbb{E}_{\mathcal{P}\left(\lfloor g_{\vartheta}^{cl}(\tilde{\boldsymbol{h}})\rceil^{k},y\right)}-\log\frac{\lfloor g_{\vartheta}^{cl}\left(\tilde{\boldsymbol{h}}\right)\rceil^{k}\times\mu_{\boldsymbol{y}}}{||\lfloor g_{\vartheta}^{cl}\left(\tilde{\boldsymbol{h}}\right)\rceil^{k}||\times||\mu_{\boldsymbol{y}}||}\\
&\approx \mathbb{E}_{\mathcal{P}\left(\lfloor\tilde{\boldsymbol{h}}\rceil^{k},y\right)}-\log\frac{\lfloor\tilde{\boldsymbol{h}}\rceil^{k}\times\mu_{\boldsymbol{y}}}{||\lfloor\tilde{\boldsymbol{h}}\rceil^{k}||\times||\mu_{\boldsymbol{y}}||}\\
&= -\mathbb{E}_{\mathcal{P}\left(\lfloor\tilde{\boldsymbol{h}}\rceil^{k},y\right)}\log\frac{\lfloor\tilde{\boldsymbol{h}}\rceil^{k}\times\mu_{\boldsymbol{y}}}{||\lfloor\tilde{\boldsymbol{h}}\rceil^{k}||\times||\mu_{\boldsymbol{y}}||},
\end{aligned}
\tag{19}
$$

which is derived by following the Bayes Rule, the definition of $\delta\left(\cdot,\cdot\right)$ in Theorem 5.1, and the fact that $g_{\vartheta}^{cl}\left(\cdot\right)$ is an MLP without amounts of deep layers so that the embedding space shift is limited. Note that $\mu_{\boldsymbol{y}}$ denotes the center of categories' feature or the corresponding classification weight of the classifier for a specific category, and $\boldsymbol{y}=\left\{\boldsymbol{y}_{l}|l\in[\![1,N^{L}]\!]\right\}$, where $N^{L}$ is the number of categories. Then, we bring Equation 19 into Equation 18 and get

$$
\begin{aligned}
&-\mathbb{E}_{\mathcal{P}\left(\lfloor\tilde{\boldsymbol{h}}\rceil^{k},y\right)}\log\frac{\lfloor\tilde{\boldsymbol{h}}\rceil^{k}\times\mu_{\boldsymbol{y}}}{||\lfloor\tilde{\boldsymbol{h}}\rceil^{k}||\times||\mu_{\boldsymbol{y}}||}\cong -\mathbb{E}_{\mathcal{P}\left(\lfloor\boldsymbol{h}\rceil^{k},y\right)}\log\frac{\lfloor\boldsymbol{h}\rceil^{k}\times\mu_{\boldsymbol{y}}}{||\lfloor\boldsymbol{h}\rceil^{k}||\times||\mu_{\boldsymbol{y}}||}\\
&\mathbb{E}_{\mathcal{P}\left(\lfloor\tilde{\boldsymbol{h}}\rceil^{k},y\right)}\log\frac{\lfloor\tilde{\boldsymbol{h}}\rceil^{k}\times\mu_{\boldsymbol{y}}}{||\lfloor\tilde{\boldsymbol{h}}\rceil^{k}||\times||\mu_{\boldsymbol{y}}||}\cong \mathbb{E}_{\mathcal{P}\left(\lfloor\boldsymbol{h}\rceil^{k},y\right)}\log\frac{\lfloor\boldsymbol{h}\rceil^{k}\times\mu_{\boldsymbol{y}}}{||\lfloor\boldsymbol{h}\rceil^{k}||\times||\mu_{\boldsymbol{y}}||}.
\end{aligned}
\tag{20}
$$

To prove Equation 20, we demonstrate an evidence example in Figure 5. As shown in the subfigure (a), the cosine error risk of the target vector $\mu_{\boldsymbol{y}}$ and the candidate vector, e.g., $\lfloor\boldsymbol{h}\rceil^{k}$, $\lfloor\tilde{\boldsymbol{h}}\rceil^{k}$, etc, is insensitive to the specific value of vectors, i.e., the cosine error risk keeps generally consistent as the values of vectors fall or vice versa. Such an assumption is supported by comparing plot (1 with plot (2 or comparing plot 3) with plot 4). Therefore, we derive

$$
\log\frac{\lfloor\tilde{\boldsymbol{h}}\rceil^{k}\times\mu_{\boldsymbol{y}}}{||\lfloor\tilde{\boldsymbol{h}}\rceil^{k}||\times||\mu_{\boldsymbol{y}}||}=\log\frac{\lfloor\boldsymbol{h}\rceil^{k}\times\mu_{\boldsymbol{y}}}{||\lfloor\boldsymbol{h}\rceil^{k}||\times||\mu_{\boldsymbol{y}}||}
\tag{21}
$$

so that

$$
\mathbb{E}_{\mathcal{P}\left(\lfloor\tilde{\boldsymbol{h}}\rceil^{k},y\right)}\log\frac{\lfloor\tilde{\boldsymbol{h}}\rceil^{k}\times\mu_{\boldsymbol{y}}}{||\lfloor\tilde{\boldsymbol{h}}\rceil^{k}||\times||\mu_{\boldsymbol{y}}||}\cong \mathbb{E}_{\mathcal{P}\left(\lfloor\boldsymbol{h}\rceil^{k},y\right)}\log\frac{\lfloor\boldsymbol{h}\rceil^{k}\times\mu_{\boldsymbol{y}}}{||\lfloor\boldsymbol{h}\rceil^{k}||\times||\mu_{\boldsymbol{y}}||}.
\tag{22}
$$

### A.2.2 Proof for the Inequality Part

To prove

$$
\Phi\left(g_{\vartheta}^{cl}\left(\tilde{\boldsymbol{h}}\right)\Big|\boldsymbol{y}\right)\leq \Phi\left(g_{\vartheta}^{cl}\left(\boldsymbol{h}\right)\Big|\boldsymbol{y}\right)
\tag{23}
$$

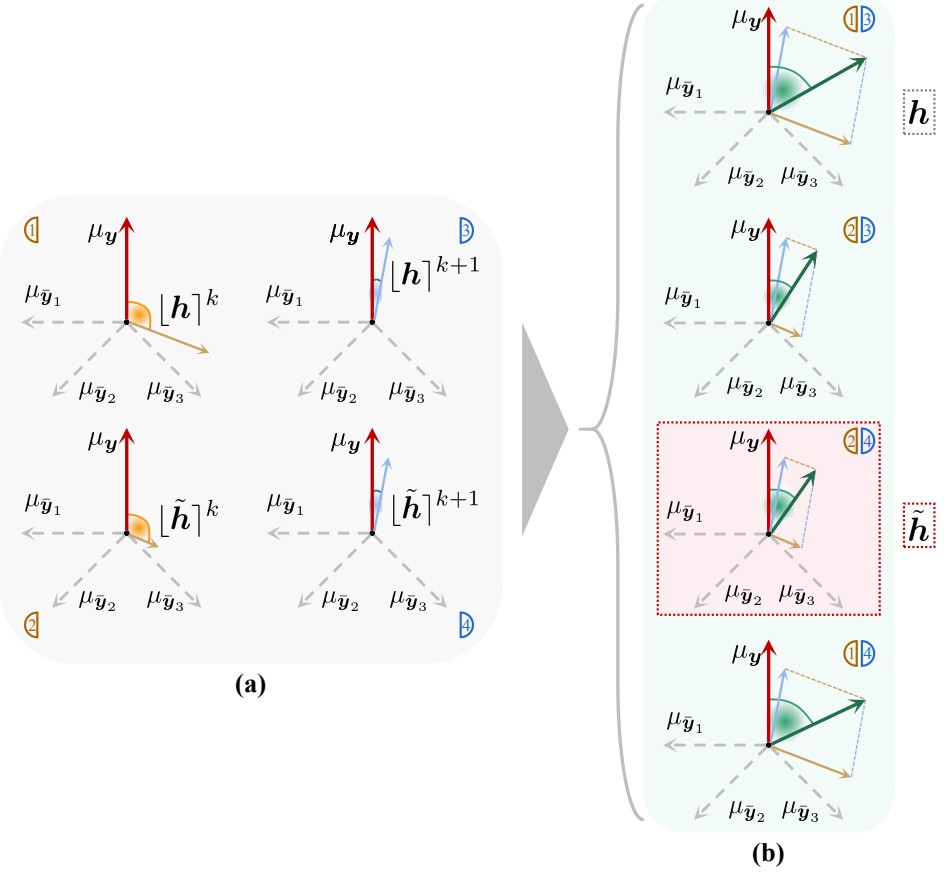

Figure 5: Evidence examples of the proofs for Theorem 5.2. (a) the visual examples of the $k$-th dimension of the unmasked representation $h$ or the masked representation $\tilde{h}$. (b) the real and hypothetical examples of $h$ or $\tilde{h}$. $\mu_y$ denotes the target category's high-dimensional feature, and $\{\mu_{\bar{y}_1}, \mu_{\bar{y}_2}, \mu_{\bar{y}_3}\}$ denote the features of other categories.

we follow the formula transformation of Equation 19 and Equation 20 to derive

$$\mathbb{E}_{\mathcal{P}(y)}\left[\mathbb{E}_{\mathcal{P}\left(g^{cl}_\vartheta(\tilde{h})|y\right)}\delta\left(g^{cl}_\vartheta\left(\tilde{h}\right), \mathbb{E}_{\mathcal{P}\left(g^{cl}_\vartheta(\tilde{h})|y\right)}\right)\right] \leq \mathbb{E}_{\mathcal{P}(y)}\left[\mathbb{E}_{\mathcal{P}\left(g^{cl}_\vartheta(h)|y\right)}\delta\left(g^{cl}_\vartheta\left(h\right), \mathbb{E}_{\mathcal{P}\left(g^{cl}_\vartheta(h)|y\right)}\right)\right]$$

$$-\mathbb{E}_{\mathcal{P}(\tilde{h},y)}\log\frac{\tilde{h}\times\mu_y}{||\tilde{h}||\times||\mu_y||} \leq -\mathbb{E}_{\mathcal{P}(h,y)}\log\frac{h\times\mu_y}{||h||\times||\mu_y||}$$

$$\mathbb{E}_{\mathcal{P}(\tilde{h},y)}\log\frac{\tilde{h}\times\mu_y}{||\tilde{h}||\times||\mu_y||} \geq \mathbb{E}_{\mathcal{P}(h,y)}\log\frac{h\times\mu_y}{||h||\times||\mu_y||}.$$

(24)

As shown in the subfigure (a) of Figure 5, all of $\lfloor h\rceil^k$, $\lfloor\tilde{h}\rceil^k$, $\lfloor h\rceil^{k+1}$, and $\lfloor\tilde{h}\rceil^{k+1}$ denote a specific dimension of the corresponding representations. Then, we reconstruct the complete representations from the dimensions, which is achieved in the subfigure (b) of Figure 5. The combinations of (2, 3) and (1, 4) are hypothetical and fake, the combination (1, 3) denotes the original and unmasked representation $h$, and (2, 4) denotes the masked representation $\tilde{h}$. We observe that although the specific value of the vector of each dimension has a thin effect on the corresponding cosine error risk, the combined representation vector is largely sensitive to such values. As demonstrated in (1, 3) and (2, 4) in the subfigure (b), the cosine error risk of $\tilde{h}$ and $\mu_y$ is apparently less than $h$ and $\mu_y$, which supports that, given a target $y$, the masked representation can indeed derive smaller conditional variance than the unmasked representation. The reason behind such a phenomenon is that, following Theorem 5.1, the self-paced dimensional mask jointly enhances the gradient effect of the dimensions containing discriminative information and reduces that of the dimensions containing the confounder by adjusting the weights of different dimensions with respect to their gradient contribution to the optimization of a specific self-supervised task, e.g., contrastive learning,

during training. Therefore, the dimensions containing discriminative information have relatively large weights, while the dimensions containing confounder information are partially *masked.*

### A.2.3 Proof for the Tighter Bounds

Being aware of proofs in Section A.2.1 and Section A.2.2, we confirm the validation of Theorem 5.2. Then, we bring Theorem 5.2 into Theorem 5.1 to derive the comparison of the lower bounds of supervised cross-entropy loss that are separately obtained by the masked representation $\tilde{h}$ and the unmasked representation $h$:

$$
\begin{aligned}
& \mathcal{L}_{contrast}\left(g_\vartheta^{cl}\left(\tilde{h}\right)\right) - \sqrt{\Phi\left(g_\vartheta^{cl}\left(\tilde{h}\right)\Big|y\right)} - \frac{1}{2}\cdot\sum_{k\in[\![1,D]\!]}\sqrt{\Phi\left(\lfloor g_\vartheta^{cl}\left(\tilde{h}\right)\rceil^k\Big|y\right)} - \mathcal{O}\left(M^{-\frac{1}{2}}\right) \\
& \geq \mathcal{L}_{contrast}\left(g_\vartheta^{cl}\left(h\right)\right) - \sqrt{\Phi\left(g_\vartheta^{cl}\left(h\right)\Big|y\right)} - \frac{1}{2}\cdot\sum_{k\in[\![1,D]\!]}\sqrt{\Phi\left(\lfloor g_\vartheta^{cl}\left(h\right)\rceil^k\Big|y\right)} - \mathcal{O}\left(M^{-\frac{1}{2}}\right).
\end{aligned}
\tag{25}
$$

Therefore, the lower bound obtained by the masked representation, i.e., MetaMask, is larger than the unmasked representation, i.e., typical contrastive learning methods. We further analyze the upper bound of supervised cross-entropy loss on downstream tasks and derive

$$
\begin{aligned}
& \mathcal{L}_{contrast}\left(g_\vartheta^{cl}\left(\tilde{h}\right)\right) + \sqrt{\Phi\left(g_\vartheta^{cl}\left(\tilde{h}\right)\Big|y\right)} + \mathcal{O}\left(M^{-\frac{1}{2}}\right) \\
& \leq \mathcal{L}_{contrast}\left(g_\vartheta^{cl}\left(h\right)\right) + \sqrt{\Phi\left(g_\vartheta^{cl}\left(h\right)\Big|y\right)} + \mathcal{O}\left(M^{-\frac{1}{2}}\right),
\end{aligned}
\tag{26}
$$

where the upper bound obtained by MetaMask is smaller than the typical contrastive learning methods. Concretely, we conclude that our approach can better bound the downstream classification risk, i.e., the upper and lower bounds of supervised cross-entropy loss obtained by MetaMask are tighter than typical contrastive learning methods.

### A.3 Discussion on the Difference between Dimensional Confounder and Curse of Dimensionality

The restricted definition of the curse of dimensionality is that given a fixed number of examples, as the dimensionality of the representation increases during training, the performance of the model on downstream tasks instead degenerates, e.g., the *Hughes phenomenon*. Recent researchers state a generalized definition of the curse of dimensionality from the perspective of over-fitting: given a fixed number of examples, the performance of the model degrades as the amount of *computation* in the network increases, e.g., the increase of representations' dimensionality or the increase of the layers of a deep network. Concretely, the curse of dimensionality is a theory to explain a specific phenomenon of over-fitting from the perspective of sample size.

However, our *dimensional confounder* is defined as a negative factor that may lead to model degradation, which is proposed from the dimensional perspective. The motivating example in Figure 2 demonstrates that even if the deep network architecture and the dimension of the representation are consistent, the dimensional confounder still causes the performance of the model to degenerate. This experimental analysis proves that the proposed dimensional confounder and the curse of dimensionality are two different terminologies. Further, to conduct the experiments in Figure 1, we fix the dimensionality of the representation and only increase the dimensionality of the features generated by the projection head, which is under the restriction of the redundancy-reduction technique [2]. The intuition behind such a behavior is that as the dimensions of the projected features increase, the information entropy contained in each dimension of the representation becomes more decoupled, and the total amount of task-relevant information entropy is limited, such that the dimensional confounder (i.e., the dimensions only containing task-irrelevant information) is gradually increasing. The experimental results demonstrate that as the dimensional confounder in the learned representation increases, the performance of the model is weakened. Such a phenomenon is consistent with the generalized definition of the curse of dimensionality, since the dimensionality of the representation is constant while the parameter (computation) of the deep network is increasing. Therefore, we conclude that the dimensional confounder can lead to the curse of dimensionality, and compared with the curse

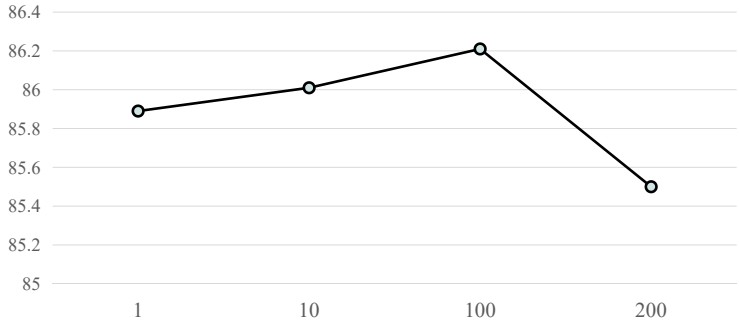

Figure 6: Comparisons of MetaMask using different settings of the hyper-parameter $\alpha$.

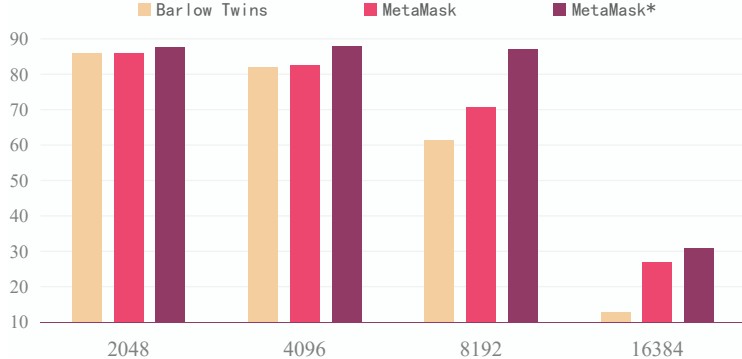

Figure 7: Further comparisons of varying the projection head's dimensionality from the range of $\{2048, 4096, 8192, 16384\}$, which are conducted on CIFAR-10 by using ResNet-18. $*$ denotes MetaMask using a trick of fixed learning rate instead of the cosine annealing strategy.

of dimensionality, the proposed dimensional confounder is a generalized issue. The comparisons in Table 1 and Table 2 prove that our MetaMask can alleviate the dimensional confounder issue, and the experiments in Figure 4 further support the superiority of MetaMask over current self-supervised methods to alleviate the curse of dimensionality issue.

## A.4 Extended Experiments

We conduct several experiments to study the intrinsic property of the proposed MetaMask.

### A.4.1 The Impact of the Hyper-Parameter $\alpha$

As demonstrated in Figure 6, we conduct comparisons of MetaMask using ResNet-18 on CIFAR-10. We follow the experimental principle in Section 6 and use KNN prediction as the evaluation approach. The comparison results show that an elaborate assignment of $\alpha$ can indeed improve the performance of MetaMask, and when $\alpha = 100$, the accuracy of MetaMask achieves the peak value. The reasons behind this curve are as follows: 1) $\mathcal{L}_{drr}$ is naturally excessively larger than $\mathcal{L}_{contrast}$ so that when $\alpha$ is not large enough, $\mathcal{L}_{contrast}$'s impact with respect to the gradient is diminished; 2) when $\alpha$ is over-large, the impact of $\mathcal{L}_{drr}$ is weakened, and the dimensional redundancy issue cannot be sufficiently addressed so that the representation may be collapsed to a trivial constant.

### A.4.2 Further Exploration of the Robustness of MetaMask

The experimental analysis demonstrated in Section 6.2 proves that MetaMask can indeed alleviate the negative impact caused by the dimensional confounder. However, MetaMask$^*$ achieves better performance than compared methods, including vanilla MetaMask. We provide a further exploration shown in Figure 7, and we observe that MetaMask$^*$ collapses abruptly at larger dimensional settings, e.g., 16384, which demonstrates our explanation of such a phenomenon in Section 6.2, i.e., the fixed learning rate trick can temporarily improve the performance of MetaMask, but this improvement is inconsistent. The results in Figure 7 further show that with or without this trick, the proposed MetaMask is robust against the dimensional confounder.

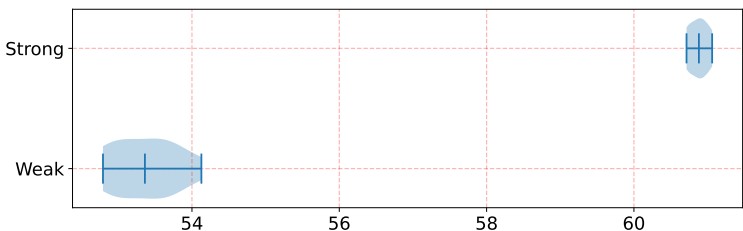

Figure 8: Comparisons of BYOL using different backbones on CIFAR100 with 5% random dimensional mask. "Weak" denotes the weak encoder *conv*, and "Strong" denotes the strong encoder *ResNet-18*.

Table 3: The complexity comparisons between MetaMask and benchmark methods on CIFAR10. Note that for fair comparisons, this experiment is based on 1 GPU of NVIDIA Tesla V100.

| Methods | Parameters | Training time cost for an epoch |
|---|---|---|
| ResNet-18 | 11.2M | - |
| SimCLR | 13M | 70s |
| Barlow Twins | 22.7M | 80s |
| SimCLR + Barlow Twins | 24.6M | 85s |
| MetaMask | 24.6M + 512 | 210s |

### A.4.3 Impacts of Encoder's Ability towards MetaMask

From the experimental results reported in Section 6, we observe that MetaMask improves the benchmark methods with weak encoders by excessively significant margins, but the improvement with strong encoders is relatively limited. Our consideration behind this phenomenon is that although representations learned by both weak and strong encoders (without our method) may contain dimensional confounders, the strong encoder can better capture semantic information so that the dimensional confounder of the learned representation is naturally less, and the useful discriminative information is much more than the representation learned by the weak encoder. [45] provide the theorem and corresponding proof to demonstrate that the contrastive loss can bound the cross-entropy loss in downstream tasks. Strong encoders better minimize the contrastive loss, and thus the representations learned by strong encoders contain more semantic information, and accordingly, fewer dimensional confounders. However, from the results reported in Table 2, our method can still improve the benchmark methods.

We further conduct experiments by imposing random dimensional masks on the learned representations for weak and strong encoders. The results are reported in Figure 8. The comparisons demonstrate that under the consistent setting of 5% random dimensional mask rate, the results of the weak encoder (conv) range from 52.79 to 54.13, and the results of the strong encoder (ResNet-18) range from 60.71 to 61.06. Note that we conduct 10 trials for each experiment to achieve unbiased results. The quality of the strong encoder's representation is apparently better, and the performance of the strong encoder is more stable (the variance of the strong encoder's results is smaller), which proves that the representation learned by the strong encoder contains fewer dimensional confounders, and each dimension contains more discriminative information. Therefore, our method improves weak encoders more than strong encoders.

### A.4.4 Training Complexity of MetaMask

To compare the training complexity of MetaMask and benchmark methods, we conduct experiments on CIFAR10 by using ResNet-18 as the backbone network. The results are demonstrated in Table 3, which shows that the parameter number used by MetaMask is close to the ablation model, i.e., SimCLR + Barlow Twins, and Barlow Twins. Compared with Barlow Twins, SimCLR + Barlow Twins only adds a MLP with several layers as the projection head for the contrasting learning of SimCLR. For MetaMask, we only add a learnable dimensional mask $\mathcal{M}$ in the network of MetaMask. Note that to decrease the time complexity of MetaMask, we create an additional parameter space to save the temporary parameters for computing second-derivatives, but such cloned parameters are not included in the computation of the network during training.

Table 4: The comparisons between MetaMask and benchmark methods on the CIFAR10 dataset by using the same total time costs. Note that for fair comparisons, this experiment is based on 1 GPU of NVIDIA Tesla V100.

| Methods | Epoch | Training time cost | Accuracy |
|---|---|---|---|
| SimCLR | 2400 | 46h | 81.75 |
| Barlow Twins | 2100 | 46h | 85.71 |
| SimCLR + Barlow Twins | 2000 | 47h | 85.79 |
| MetaMask | 800 | 46h | 86.01 |

Table 5: The comparisons of Barlow Twins using different dropout ratios on CIFAR10 with ResNet-18.

| Dropout ratio | Dropout shared between views | Accuracy |
|---|---|---|
| 0 | No | 85.72 |
| 0.01 | No | 85.76 |
| 0.05 | No | 86.11 |
| 0.1 | No | 85.72 |
| 0.2 | No | 85.13 |
| 0.5 | No | 81.54 |
| 0.05 | Yes | 31.17 |
| 0.1 | Yes | 37.15 |
| 0.5 | Yes | 31.62 |

For the time complexity, due to the learning paradigm of meta-learning (second-derivatives technique), MetaMask has larger time complexity than benchmark methods (including the ablation model SimCLR + Barlow Twins). We further conduct experiments to evaluate the performance of the compared methods using similar total training time costs. The results are reported in Table 4, which demonstrates that MetaMask still achieves the best performance, and MetaMask can also beat the ablation model SimCLR + Barlow Twins. This proves that although the time complexity of MetaMask is a little bit high, the improvement of MetaMask is consistent and solid. We also include these results and the corresponding analysis in Appendix of the rebuttal revised version.

### A.4.5   Exploration of Barlow Twins using the Dropout Trick

We apply dropout to randomly set the features generated by the backbone network to 0 with a given probability. We use two different settings: 1) features from different views are randomly set to 0 independently with 6 probabilities, including 0.01, 0.05, 0.1, 0.2, and 0.5, which is shown as the setting of dropout shared between views is "No"; 2) for the same sample, we set the same channels of the features from different views to 0. In detail, we let the features from the first view pass the dropout layer, and then record the channels which are set to 0. Finally, we set the same feature channels of the second view to 0 and multiply the features with $1/(1-p)$, where "p" refers to the probability of dropout. In this case, we use three probabilities: 0.05, 0.1, and 0.5. This setting is shown as "Yes" for the dropout shared between views. The results are reported in Table 5. We observe that models trained by following the second setting are collapsed, and these models far underperform MetaMask. For the first setting, the performance trend is in accordance with our expectation, using several well-chosen dropout ratios can improve the performance of Barlow Twins to a certain extent, but the improvement is inconsistent. Furthermore, even the best model of Barlow Twins using the dropout trick cannot achieve comparable performance to MetaMask, since as shown in Table 2, MetaMask achieves 87.53 on CIFAR10 with ResNet-18.

### A.4.6   Performing MetaMask on ImageNet with ResNet-50

Training one trial of MetaMask on ImageNet is excessively time-consuming for the adopted server. It is hard to impose sufficient hyperparameter tuning experiments, because tuning parameters on the validation set and then retraining is time-consuming. For the current version, we principally adopt the hyperparameter of MetaMask on ImageNet-200 with ResNet-18 for the experiments on ImageNet with ResNet-50. We report the results on Table 6, which demonstrates that MetaMask achieves the

Table 6: Top-1 and top-5 accuracies (%) under linear evaluation on ImageNet with ResNet-50 encoder. MetaMask is performed based on Barlow Twins and SimCLR. Top-3 best methods are underlined.

| Method | Accuracy | |
|---|---|---|
| | Top-1 | Top-5 |
| Supervised | 76.5 | - |
| MoCo | 60.6 | - |
| SimCLR | 69.3 | 89.0 |
| SimSiam | 71.3 | - |
| BYOL | 74.3 | 91.6 |
| Barlow Twins | 73.2 | 91.0 |
| MetaMask | 73.9 | 91.4 |

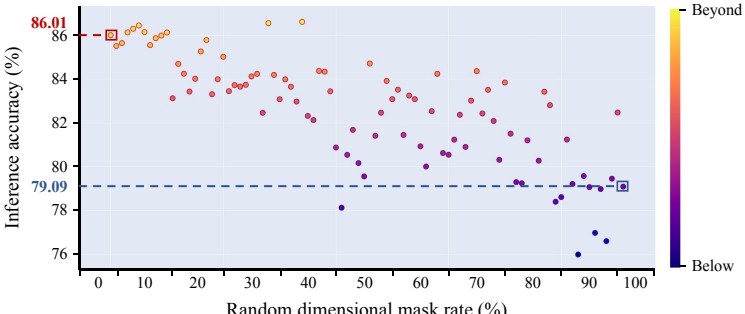

Figure 9: Experimental scatter diagrams obtained by MetaMask with randomly masked dimensions on CIFAR-10 with ResNet-18. We collect the final dimensional weight matrix $\mathcal{M}$ and then choose dimensions with weights below average as the masked dimensions. These dimensions are considered to be associated with the dimensional confounders, and we impose random dimensional masks on these dimensions. For fair comparisons, 10 trials are conducted per mask rate, except for 0% and 100% mask rates, and every single point denotes a trial. The trial boxed by *red lines* presents the performance achieved by the unmasked representation of MetaMask, and the trial boxed by *blue lines* presents the performance achieved by MetaMask's representation where all dimensions containing confounders are masked.

top-3 best performance. Note that the major results are reported by [2], and the implementation of MetaMask is based on SimCLR and Barlow Twins. For comparisons with the main baselines, MetaMask beats SimCLR by 4.6 on top-1 accuracy and 2.4 on top-5 accuracy, and MetaMask beats Barlow Twins by 0.7 on top-1 accuracy and 0.4 on top-5 accuracy. The improvement of MetaMask is in accordance with our observation in Appendix A.4.3. Furthermore, ImageNet-200 is a truncated dataset of ImageNet, so the domain shift between these datasets is slight. The comparisons in Table 1, and Table 2 demonstrate that the proposed MetaMask can still improve the benchmark methods in large-scale datasets.

### A.4.7 Exploration of Confounders Contained by Masked Dimensions

MetaMask trains $\mathcal{M}$ by adopting a meta-learning-based training approach, which ensures that $\mathcal{M}$ can partially mask the "gradient contributions" of dimensions containing task-irrelevant information and further promote the encoder to focus on learning task-relevant information. So, MetaMask only performs the gradient mask during training instead physically masking dimensions in the test. We provide theoretical explanation and proofs in Appendix A.1 and Appendix A.2. The reason behind our behavior is that even dimensions that contain dimensional confounders are also possible to contain discriminative information so that lowering the gradient contribution of such dimensions can not only prevent the over-interference of the dimensional confounders on the representation learning but also preserve the acquisition of the information of these dimensions. Accordingly, the foundational idea behind self-supervised learning is to learn "general" representation that can be generalized to various tasks. MetaMask introduces a meta-learning-based approach to train the dimensional mask $\mathcal{M}$ with respect to improving the performance of contrastive learning. However, the theorems,

proposed by [45], only prove that the contrastive learning objective is associated with the downstream classification task, while there is no theoretical evidence to demonstrate the connection between contrastive learning objective and other downstream tasks. Therefore, we consider not directly masking the dimensions containing dimensional confounders in the test.

Furthermore, we conduct experiments to explore the performance of the variant that directly masks these dimensions in the test, which is demonstrated in Figure 9. For the exploration of our masking scheme and its variants, we conduct experiments as follows: after training, we collect the final dimensional weight matrix $\mathcal{M}$ and then choose dimensions with weights below average as the masked dimensions. These dimensions are considered to be associated with dimensional confounders. To prove whether these dimensions have confounders, we perform random dimensional masking to these dimensions, and when the masking rate is 100%, the model turns to the variant that directly masks all these dimensions in the test. The experiments are based on SimCLR + MetaMask. Note that we conduct 10 trials per mask rate (except for 0% and 100% mask rates) for fair comparisons. We observe that the original MetaMask (i.e., mask rate is 0%) achieves the best performance on average, and MetaMask outperforms the variant masking the dimensions with confounders by a significant margin, which proves that our proposed approach, i.e., masking the "gradient contributions" of dimensions in the training is more effective than the compared approach, i.e., directly masking dimensions in the test. While, several trials with specific mask rates demonstrate better performance than MetaMask, which proves that the dimensions filtered by MetaMask indeed contain dimensional confounders. Additionally, observing the results reported in Figure 2 and the results in Figure 9, we find that the results achieved by the proposed variants are better than Barlow Twins with random dimensional masks on average, which can further prove the filtered dimensions containing confounders and that MetaMask indeed assigns lower gradient weights to the dimensions containing confounders.

Additionally, we conduct experiments to directly impose a linear probe on the whole masked or unmasked features. Note that the definition of masked features is mentioned above, and the experiments are conducted on CIFAR10 with ResNet-18. As shown in Figure 9, a linear probe is solely imposed on the whole "unmasked" features (without "masked" features), which is the same as the 100% mask rate variant for the "masked" features above, and the result is 79.09. We provide the corresponding reasons and analyses above. Additionally, such results may be due to the masking scheme for these experiments, i.e., collecting the final dimensional weight matrix $\mathcal{M}$ and then masking the dimensions with weights below average, which is dramatically different from MetaMask's behavior, and this scheme is only used for this exploration. The result achieved by a linear probe on the whole "masked" features is 56.30, which demonstrates our statement that dimensions containing dimensional confounders are also possible to contain discriminative information, because the achieved accuracy is not under 10. The result also proves that MetaMask can indeed assign lower gradient weights to the dimensions containing confounders, since such a model far underperforms MetaMask (86.01).

### A.4.8  Further Ablation Study of MetaMask

Barlow Twins handles dimensional redundancy but suffers dimensional confounders. MetaMask mitigates dimensional confounders by learning and applying a dimensional mask. We conduct experiments to demonstrate our statement, and the results are reported in Figure 10. For the experiments shown in Figure 10 (a), we demonstrate the effectiveness of the proposed dimensional mask $\mathcal{M}$ and the corresponding meta-learning-based training paradigm by directly removing such approaches. The results show that the sole w/o ML only improves Barlow Twins by 0.04, while MetaMask can improve BT by 0.29, which proves the proposed $\mathcal{M}$ is pivotal to the performance promotion. In Figure 10 (b), to verify whether there would be performance gain only from alleviating dimensional confounders without $\mathcal{L}_{drr}$, we evaluate the performance of w/o drr and MetaMask. We observe that for the experiments with ResNet-18, w/o drr (without Barlow Twins) improves SimCLR by 0.2 but cannot reach the performance of BT, and MetaMask can improve both SimCLR and BT by 4.28 and 0.29, respectively. For the experiments with conv, w/o drr improves SimCLR by 1.03 and also outperforms BT by 4.08, and MetaMask improves both SimCLR and BT by 2.27 and 6.22, respectively. Concretely, the performance of w/o drr is related to the performance of SimCLR, and it can always improve SimCLR but underperform the complete MetaMask. MetaMask has consistent best performance by using different encoders.

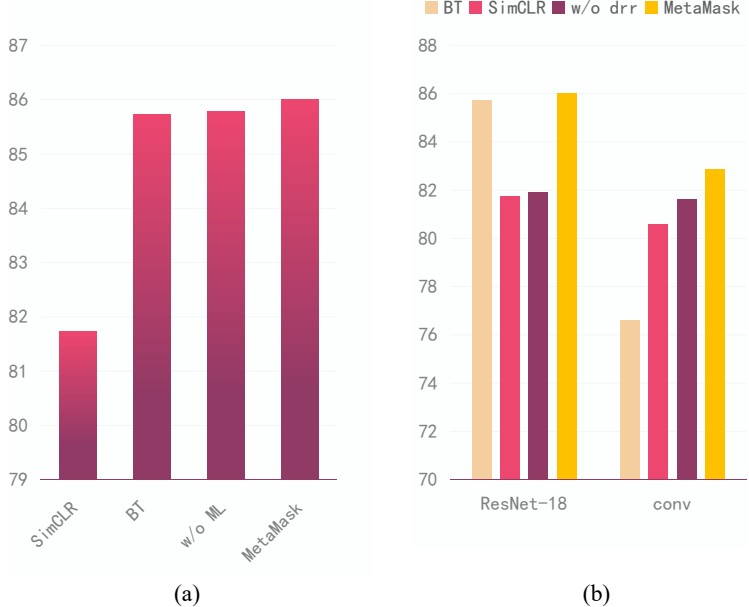

(a)                  (b)

Figure 10: Ablation studies obtained by MetaMask on CIFAR-10 with ResNet-18 and conv encoders, where "BT" denotes Barlow Twins, "w/o ML" denotes the ablation model without the proposed $\mathcal{M}$ and the corresponding meta-learning-based learning paradigm, i.e., SimCLR + Barlow Twins, and "w/o drr" denotes the ablation model without the dimensional redundancy reduction loss $\mathcal{L}_{drr}$. (a) We conduct experiments to prove the effectiveness of the proposed dimensional mask $\mathcal{M}$ on CIFAR10 with ResNet-18, and the results show that only w/o ML (SimCLR + Barlow Twins) can not improve the performance of the model by a significant margin, and the proposed approach is crucial to the performance improvement. (b) We conduct experiments on CIFAR10 with ResNet-18 and conv, respectively. The results further prove the effectiveness of the proposed $\mathcal{M}$ and $\mathcal{L}_{drr}$.

We consider that $\mathcal{L}_{drr}$ (Barlow Twins loss) could exacerbate dimensional confounders, but as our discussion in Appendix A.4.7, i.e., dimensions containing confounders are also possible to contain discriminative information, more dimensions with confounders due to $\mathcal{L}_{drr}$ may also carry more discriminative information. Likewise, the model without $\mathcal{L}_{drr}$ may generate representations with over-redundant dimensions so that the total amount of available discriminative information will decrease. However, roughly using the representations with complex dimensional information (without $\mathcal{L}_{drr}$) may result in insufficient discriminative information mining, e.g., w/o ML can only outperform SimCLR by a limited margin. Our proposed MetaMask effectively avoids the appearance of such an undesired phenomenon by leveraging $\mathcal{M}$ and the corresponding meta-learning-based training paradigm, which is supported by the empirical results.