# OpenReview forum: "MetaMask: Revisiting Dimensional Confounder for Self-Supervised Learning"
_NeurIPS.cc/2022/Conference — NeurIPS 2022 Accept_

### Official Review · Reviewer_FKJ2 · 2022-07-04

**Rating:** 5
**Confidence:** 4
**Soundness:** 2 fair
**Presentation:** 1 poor
**Contribution:** 2 fair

**Summary:**

This paper proposed a hybrid joint-embedding framework, which adds learnable masks on representations and applies both contrastive loss and redundancy reduction loss. This mask is updated when fixing the rest of the network.
The experiments show that the proposed method improves most of the recent frameworks (BarlowTwins, SimCLR, SwAV, BYOL, etc.) on various datasets (CIFAR-10, CIFAR-100,  STL-10, IN-200)

**Questions:**

1. What's the reason behind MetaMask's collapse for large embedding dimensions? This is not observed in standard BarlowTwins.

2. Why is this considered "meta"? It's basically second-order optimization.


**Strengths And Weaknesses:**

Strengths
1. The authors point out that joint-embedding methods will learn harmful features. This is a widely ignored problem in the self-supervised learning method.
2. Experiment results show consistent improvement of the proposed method overall frameworks on various datasets.
3. The second-order optimization on the mask is supported by theoretical proof.


Weakness
1. The masks are trainable means that in the end, they are fixed and will ignore several features known as confounders. However, this needs to be verified via ablation experiments. For example, a linear probe on the whole unmasked features should give lower accuracy.
2. The experiments are not convincing due to a lack of fair comparison. The authors use a very customized setting, e.g., AlexNet on various joint-embedding frameworks. No hyperparameter tuning is conducted on these models.
3. Second-order optimization creates significant computational overhead. The authors need to show how much more training time is needed for each model.
4. No experiment details.


====== post rebuttal comments =====

I've carefully read the authors' responses. All my major concerns are addressed. Though the AlexNet setting on 1 GPU is not convincing enough, the fair amount of controlled experiments in the paper convinces me of the effectiveness of the proposed idea.

I've increased my score.

---

> ### Author Response · Authors · 2022-08-02
> **Added Experiments and Analyses of Masking Dimensions that Contains Confounders, and Computational Costs, Clarifications of the Learning Paradigm, the Implementations of Compared Methods, Descriptions of experimental details, and Discussions on the Reason behind MetaMask's Collapse for Large Embedding Dimensions (3/3)**
>
> Q4: No experiment details.
>
> A4: We provide the implementation details in the code of the supplementary file. For example, for the experiments on CIFAR10, we use ResNet-18 as the backbone network. We train the network for 800 epochs with the batch size of 512. In particular, for the experiments of BYOL, we adopt Adam to optimize the network with the learning rate of 5e-4. We leverage SGD as the optimizer for all other methods and set the learning rate 1e-3. The cosine annealing strategy is used when updating the learning rate. We adopt the principle experimental settings by following the experiments of the corresponding implementations of benchmark methods, e.g., Barlow Twins, SimCLR, BYOL, etc.
>
> Q5: What's the reason behind MetaMask's collapse for large embedding dimensions? This is not observed in standard BarlowTwins.
>
> A5: Thanks for your comments. We conduct the motivating experiments in Figure 1 (Page 2) by adopting the official implementation of Barlow Twins, and thus the phenomenon of the model collapsing for large embedding dimensions are shared between standard Barlow Twins and MetaMask. We consider the reason why our exploration sharply contrasts with Barlow Twins are: 1) Barlow Twins does not take experiments on small-scale datasets, e.g., CIFAR10; 2) the dimensionality taken by Barlow Twins is not large enough. In particular, the projection head dimensionality range of Barlow Twins is 16 to 16384, while that in our paper is 512 to 40960, and the results of lower dimensionality are consistent between Barlow Twins and ours.
>
> Furthermore, in further exploration, we propose that such a model collapsing phenomenon is due to two reasons: 1) the proposed dimensional confounder; 2) the curse of dimensionality. And we elaborate on the difference and our intuitive analysis in Appendix A.3. The experiments, shown in Figure 4 (Page 9) and Figure 7 (Page 19), demonstrate that MetaMask can yield significant performance boosts against the mode collapse.
>
> Q6: Why is this considered "meta"? It's basically second-order optimization.
>
> A6: Thanks for the comment. The original idea of meta-learning is to learn an initial model (with parameters) that can quickly learn discriminative information by a few epochs of training, and the knowledge of such an initial model is considered the ''meta'' knowledge. This goal is achieved by adopting a second-derivative technique to train the initial model. The intuition on this behavior is to train the initial model with respect to ''improving'' the performance of the trained model on various tasks. [40] and our method holds the shared intuition of meta-learning. Specifically, [40] proposes the meta auxiliary learning to train the auxiliary learning model with respect to ''improving'' the self-supervised learning, and our method proposes to train the learnable dimensional mask matrix $\mathcal{M}$ with respect to ''improving'' the contrastive learning. Therefore, following the naming principle, we call our second-order optimization alike training paradigm as ''meta''.
>
> References:
>
> [5] Yonglong Tian, Dilip Krishnan, and Phillip Isola. Contrastive multiview coding. arXiv preprint arXiv:1906.05849, 2019.
>
> [40] S. Liu, Andrew J Davison, and E. Johns. Self-supervised generalisation with meta auxiliary learning. 2019.

---

> ### Author Response · Authors · 2022-08-02
> **Added Experiments and Analyses of Masking Dimensions that Contains Confounders, and Computational Costs, Clarifications of the Learning Paradigm, the Implementations of Compared Methods, Descriptions of experimental details, and Discussions on the Reason behind MetaMask's Collapse for Large Embedding Dimensions (2/3)**
>
> Additionally, following the suggestion of the reviewer, we conduct experiments to directly impose a linear probe on the whole masked or unmasked features. Note that the definition of masked features is mentioned above, and the experiments are conducted on CIFAR10 with ResNet-18. As shown in Appendix A.4.7, a linear probe is solely imposed on the whole ''unmasked'' features (without ''masked'' features), which is the same as the 100% mask rate variant for the ''masked’’ features above, and the result is 79.09. We provide the corresponding reasons and analyses above. Additionally, such results may be due to the masking scheme for these experiments, i.e., collecting the final dimensional weight matrix $\mathcal{M}$ and then masking dimensions with weights below average, which is dramatically different from MetaMask's behavior, and this scheme is only used for this exploration. The result achieved by a linear probe on the whole ''masked'' features is 56.30, which demonstrates our consideration that dimensions containing dimensional confounders are also possible to contain discriminative information, because the achieved accuracy is not under 10. The result also proves that MetaMask can indeed assign lower gradient weights to the dimensions containing confounders, since such a model far underperforms MetaMask (86.01).
>
> Q2: The experiments are not convincing due to a lack of fair comparison. The authors use a very customized setting, e.g., AlexNet on various joint-embedding frameworks. No hyperparameter tuning is conducted on these models.
>
> A2: Thanks for the comments. For the experiments using different layers of AlexNet, we follow the experimental setting of [5], and part of the experimental results refer to the corresponding papers, where the hyper-parameters are tuned for different compared models. Additionally, we also provide experiments by using ResNet-18 as the backbone network, and the results also demonstrate the effectiveness of MetaMask. All reimplementations are built by following the official implementations.
>
> Table 1. The complexity comparisons between MetaMask and benchmark methods on the CIFAR10 dataset. Note that for fair comparisons, this experiment is based on 1 GPU of NVIDIA Tesla V100.
> |Methods | Parameters | Training time cost for an epoch |
> | :------| :----| :----|
> | ResNet-18 | 11.2M | - |
> | SimCLR | 13M | 70s |
> | Barlow Twins | 22.7M | 80s |
> | SimCLR + Barlow Twins | 24.6M | 85s |
> | MetaMask | 24.6M + 512 | 210s |
>
> Table 2. The comparisons between MetaMask and benchmark methods on the CIFAR10 dataset by using the same total time costs. Note that for fair comparisons, this experiment is based on 1 GPU of NVIDIA Tesla V100.
> |Methods | Epoch | Training time cost | Accuracy |
> | :------| :----| :----| :----|
> | SimCLR | 2400 | 46h | 81.75 |
> | Barlow Twins | 2100 | 46h | 85.71 |
> | SimCLR + Barlow Twins | 2000 | 47h | 85.79 |
> | MetaMask |800 | 46h | 86.01 |
>
> Q3: Second-order optimization creates significant computational overhead. The authors need to show how much more training time is needed for each model.
>
> A3: To compare the training complexity of MetaMask and benchmark methods, we conduct experiments on CIFAR10 by using ResNet-18 as the backbone network. The results are demonstrated in Table 1, which shows that the parameter number used by MetaMask is close to the ablation model, i.e., SimCLR + Barlow Twins, and Barlow Twins. Compared with Barlow Twins, SimCLR + Barlow Twins only adds a MLP with several layers as the projection head for the contrasting learning of SimCLR. Comparing SimCLR + Barlow Twins and MetaMask, we only add a learnable dimensional mask $\mathcal{M}$ in the network of MetaMask. Note that to decrease the time complexity of MetaMask, we create an additional parameter space to save the temporary parameters for computing second-derivatives, but such cloned parameters do not participate in the calculation of the network during training.
>
> For the time complexity, due to the learning paradigm of meta-learning (second-derivatives technique), MetaMask has larger time complexity than benchmark methods (including the ablation model SimCLR + Barlow Twins). While, we further conduct experiments to evaluate the performance of the compared methods using similar total training time costs. The results are reported in Table 2, which demonstrates that MetaMask still achieves the best performance, and MetaMask can also beat the ablation model SimCLR + Barlow Twins. This proves that although the time complexity of MetaMask is a little bit high, the improvement of MetaMask is consistent and solid. We also include these results and the corresponding analysis in Appendix A.4.4 of the rebuttal revised version. Thank you for this suggestion.

---

> ### Author Response · Authors · 2022-08-02
> **Added Experiments and Analyses of Masking Dimensions that Contains Confounders, and Computational Costs, Clarifications of the Learning Paradigm, the Implementations of Compared Methods, Descriptions of experimental details, and Discussions on the Reason behind MetaMask's Collapse for Large Embedding Dimensions (1/3)**
>
> We thank the reviewer for the valuable comments and constructive suggestions. We are encouraged that the reviewer found this work is novel, the explored problem is widely ignored, and the theoretical proof is integrated. The mentioned issues are addressed as follows:
>
> Q1: The masks are trainable means that in the end, they are fixed and will ignore several features known as confounders. However, this needs to be verified via ablation experiments. For example, a linear probe on the whole unmasked features should give lower accuracy.
>
> A1: Thanks for the suggestions! First, we clarify that MetaMask trains $\mathcal{M}$ by adopting a meta-learning-based training approach, which ensures that $\mathcal{M}$ can partially mask the ''gradient contributions'' of dimensions containing task-irrelevant information and further promote the encoder to focus on learning task-relevant information. So, MetaMask only performs the gradient mask during training instead physically masking dimensions in the test. We provide theoretical explanation and proofs in Appendix A.1 and A.2. The reason behind our behavior (adjusting the gradient weight of each dimension in training instead of directly masking these dimensions in the test) is that even dimensions that contain dimensional confounders are also possible to contain discriminative information so that lowering the gradient contribution of such dimensions can not only prevent the over-interference of the dimension confounders on the representation learning but also preserve the acquisition of the information of these dimensions. Accordingly, the foundational idea behind self-supervised learning is to learn ''general'' representation that can be generalized to various tasks. In MetaMask, we introduce a meta-learning-based approach to train the dimensional mask $\mathcal{M}$ with respect to improving the performance of contrastive learning. However, the theorems, proposed by [41] and us (in Section 5, Appendix A.1, and Appendix A.2), only prove that the contrastive learning objective is associated with the downstream classification task, while there is no evidence to demonstrate the connections between contrastive learning objective and other downstream tasks. Therefore, we consider not directly masking the dimensions containing dimensional confounders in the test.
>
> Furthermore, we conduct experiments to explore the performance of the variant that directly masks these dimensions in the test, which is demonstrated in Appendix A.4.7 of the rebuttal revised version. For the exploration of our masking scheme and its variants, we conduct experiments as follows: after training, we collect the final dimensional weight matrix $\mathcal{M}$ and then choose dimensions with weights below average as the masked dimensions. These dimensions are considered to be associated with dimensional confounders. To prove whether these dimensions have confounders, we perform random dimensional masking to these dimensions, and when the masking rate is 100%, the model turns to the variant that directly masks all these dimensions in the test. The experiments are based on SimCLR + MetaMask. Note that we conduct 10 trials per mask rate (except for 0% and 100% mask rates) for fair comparisons. We observe that the original MetaMask (i.e., mask rate is 0%) achieves the best performance on average, and MetaMask outperforms the variant masking the dimensions with confounders by a significant margin, which proves that our proposed approach, i.e., masking the ''gradient contributions'' of dimensions in the training is more effective than the compared approach, i.e., directly masking dimensions in the test. While, several trials with specific mask rates demonstrate better performance than MetaMask, which proves that the dimensions filtered by MetaMask indeed contain dimensional confounders. Additionally, observing the results reported in Figure 2 (Page 3) and the results in Appendix A.4.7, we find that the results achieved by the proposed variants are better than Barlow Twins with random dimensional masks on average, which can further prove the filtered dimensions containing confounders and that MetaMask indeed assigns lower gradient weights to the dimensions containing confounders.

---

> ### Author Response · Authors · 2022-08-09
> **Final response to the reviewer**
>
> Since the discussion phase is closing soon, this response could be our last chance to discuss with the reviewer. We would be grateful for the careful review and constructive suggestions of the reviewer. We hope our rebuttals may make our intuition behind this paper more understandable and clearer. Following the reviewer's suggestions, we improve our paper from multiple aspects, and the we expect that the revised manuscript can address your concerns.

---

### Official Review · Reviewer_ZgDJ · 2022-07-08

**Rating:** 6
**Confidence:** 4
**Soundness:** 3 good
**Presentation:** 3 good
**Contribution:** 3 good

**Summary:**

The authors propose a self-supervised learning scheme with a mask on top of the contrastive learning and the Barlow-twin method to perform dimensional reduction and confounder elimination. The concept is to use a meta-learning approach to seek a mask deleting redundant features while optimizing the feature representation and model parameters on the fly. The training loss is based on a combination of contrastive and Barlow-twin.

**Questions:**

I think the improvement is good, but I want to know that this meta-learning type is more essential than just optimizing a loss of Barlow-twin and contrastive loss. Also, the theoretical point that mask includes identity seemingly makes the theorem vacuous, and I wonder if the authors can respond to my concern.

**Limitations:**

Yes

**Strengths And Weaknesses:**

Strengths:

The proposed method seems easy to implement based on existing works and achieve good performance improvement. The method has been validated with theory.

Weakness:

The ablation study is insufficient, as the method uses both Barlow-twin and contrastive loss. Still, it's unclear that this meta-learning alike optimization scheme is essential compared with training on a loss combination of Barlow-twin and contrastive loss. The comparisons have only been made against either one of them. Also, in Figure 1, the performance improvement by the random mask is not significant enough to persuade me that the feature redundancy is sufficiently essential (only within 0.1% improvement on ImageNet). Finally, why is the performance not tested on full ImageNet while the motivation figure is?

Theoretically, the authors show that the conditional variance can be reduced with the optimal mask. Still, since the identity matrix (no mask) is also a special case of masks, this is a seemingly trivial statement.

---

> ### Author Response · Authors · 2022-08-02
> **Ablation Study on the Two Parts of MetaMask, Added Experimental Analyses of Impacts of Encoders’ Ability on MetaMask and the Comparisons on ImageNet, and Clarifications of the Theoretical Statement (3/3)**
>
> Q3: Theoretically, the authors show that the conditional variance can be reduced with the optimal mask. Still, since the identity matrix (no mask) is also a special case of masks, this is a seemingly trivial statement.
>
> A3: Thanks for your comments! The identity matrix (without mask) aims to reduce the conditional variance of each specific dimension, i.e., the same dimension of different views of a sample contains similar information. Although such an approach can reduce the conditional variance of the representations, the theoretical proof to demonstrate that such an approach can reduce the conditional variance (conditional on the ''label'' not the dimensions themselves) of the learned representation in hidden space is insufficient, because Theorem 4.2 [41] only proves that the conditional variance (conditional on the ''label'') is related to the contrastive objective, and the theoretical evidences proving the connection between such a conditional variance and the identity matrix is deficient. Our Theorem 5.1 (Page 8) further proves that benefiting from the proposed learning paradigm (meta-learning-based approach), MetaMask can further reduce the variance of the learned representation conditional on the label. We provide corresponding proofs in Appendix A.2. Furthermore, the empirical evidences, in Table 1 (Page 8) and Table 2 (Page 9), demonstrate that MetaMask has better performance than benchmark methods, including Barlow Twins (only using the identity matrix), so compared with other methods, given the label, the conditional variance of the representation learned by MetaMask is reduced.
>
> Q4: I think the improvement is good, but I want to know that this meta-learning type is more essential than just optimizing a loss of Barlow-twin and contrastive loss. Also, the theoretical point that mask includes identity seemingly makes the theorem vacuous, and I wonder if the authors can respond to my concern.
>
> A4: Thanks for your careful review! We conduct the corresponding exploration in Appendix A.4.8, and the analysis is mentioned in A1. For the issue of theoretical points, we clarify the significance and explanations in A3.
>
> References:
>
> [14] Jure Zbontar, Li Jing, Ishan Misra, Yann LeCun, and Stéphane Deny. Barlow twins: Self-supervised learning via redundancy reduction. In Proceedings of the 38th International Conference on Machine Learning, ICML 2021, 18-24 July 2021, Virtual Event, Proceedings of
> Machine Learning Research. PMLR, 2021.
>
> [41] Yifei Wang, Qi Zhang, Yisen Wang, Jiansheng Yang, and Zhouchen Lin. Chaos is a ladder: A new theoretical understanding of contrastive learning via augmentation overlap. arXiv preprint arXiv:2203.13457, 2022.

---

> ### Author Response · Authors · 2022-08-02
> **Ablation Study on the Two Parts of MetaMask, Added Experimental Analyses of Impacts of Encoders’ Ability on MetaMask and the Comparisons on ImageNet, and Clarifications of the Theoretical Statement (2/3)**
>
> Table 1. Top-1 and top-5 accuracies (in %) under linear evaluation on ImageNet with ResNet-50 encoder. MetaMask is performed based on Barlow Twins and SimCLR.
> | Model | Top-1 | Top-5 |
> | :------| :----| :----|
> | Supervised | 76.5 | - |
> | MoCo | 60.6 | - |
> | SimCLR | 69.3 | 89.0 |
> | SimSiam | 71.3 | - |
> | BYOL | 74.3 | 91.6 |
> | Barlow Twins | 73.2 | 91.0 |
> | MetaMask | 73.9 | 91.4 |
>
> Q2: Also, in Figure 1, the performance improvement by the random mask is not significant enough to persuade me that the feature redundancy is sufficiently essential (only within 0.1% improvement on ImageNet). Finally, why is the performance not tested on full ImageNet while the motivation figure is?
>
> A2: Thanks for your careful review! The experiments conducted on ImageNet are based on ResNet-50, while the experiments conducted on other benchmark datasets are based on ResNet-18 or conv or fc. Throughout our exploration, we observe a trend: MetaMask improves the benchmark methods with weak encoders by excessively significant margins, but the improvement with strong encoders is relatively limited. Our consideration behind this phenomenon is that although representations learned by both weak and strong encoders (without our method) may contain dimensional confounder, the strong encoder can better capture semantic information so that the dimensional confounder of the learned representation is naturally less, and the useful discriminative information is much more the representation learned by the weak encoder. [41] provided the theorem and corresponding proof to demonstrate that the contrastive loss can bound the cross-entropy loss in downstream tasks. Therefore, strong encoders better minimize the contrastive loss, which means the representations learned by strong encoders contain more semantic information, and accordingly, fewer dimensional confounders. However, from the results reported in Table 2 (Page 9), our method can still improve the benchmark methods.
>
> We further conduct experiments by imposing random dimensional masks on the learned representations for weak and strong encoders. The results are reported in Appendix A.4.3 of the rebuttal revised version. The comparisons demonstrate that under the consistent setting of the 5% random dimensional mask rate, the results of the weak encoder (conv) range from 52.79 to 54.13, and the results of the strong encoder (ResNet-18) range from 60.71 to 61.06. Note that we conduct 10 trials for each experiment to achieve unbiased results. The quality of the strong encoder's representation is better, and the performance of the strong encoder is more stable (the variance of the strong encoder's results is smaller), which proves that the representation learned by the strong encoder contains less dimensional confounder, and each dimension contains more discriminative information, and our method can improve weak encoders more than strong encoders.
>
> ImageNet, as a large-scale dataset, has enough training data to improve the encoder to learn discriminative information, and the backbone encoder used on ImageNet is stronger, i.e., ResNet-50. Therefore, in Figure 1, compared with the improvement on CIFAR10, the performance improvement by the random mask on ImageNet is less significant.
>
> Training one trial of MetaMask on ImageNet takes about 14 days for our server. It is hard for us to impose sufficient hyperparameter tuning experiments, because tuning parameters on the validation set and then retraining is time-consuming. For the current version, we principally adopt the hyperparameter of MetaMask on ImageNet-200 with ResNet-18 for the experiments on ImageNet with ResNet-50. We report the results in Table 1, which demonstrates that MetaMask achieves the top-2 best performance. Note that the major results are reported by [14], and the implementation of MetaMask is based on SimCLR and Barlow Twins. For comparisons with the main baselines, MetaMask beats SimCLR by 4.6 on top-1 accuracy and 2.4 on top-5 accuracy, and MetaMask beats Barlow Twins by 0.7 on top-1 accuracy and 0.4 on top-5 accuracy. The improvement of MetaMask is in accordance with our observation above. Furthermore, ImageNet-200 is a truncated dataset of ImageNet, so the domain shift between these datasets is slight. The comparisons in Table 1 (Page 8), and Table 2 (Page 9) demonstrate that the proposed MetaMask can still improve the benchmark methods in large-scale datasets. We add the newly achieved results and corresponding analysis in Appendix A.4.6 of the rebuttal revised version.

---

> ### Author Response · Authors · 2022-08-02
> **Ablation Study on the Two Parts of MetaMask, Added Experimental Analyses of Impacts of Encoders’ Ability on MetaMask and the Comparisons on ImageNet, and Clarifications of the Theoretical Statement (1/3)**
>
> We thank the reviewer for the valuable comments and constructive suggestions. We are encouraged that the reviewer found that this work is novel and technically sound. The mentioned issues are addressed as follows:
>
> Q1: The ablation study is insufficient, as the method uses both Barlow-twin and contrastive loss. Still, it's unclear that this meta-learning alike optimization scheme is essential compared with training on a loss combination of Barlow-twin and contrastive loss. The comparisons have only been made against either one of them.
>
> A1: Thanks for the suggestions! We added the corresponding analysis in Appendix A.4.8 of the rebuttal revised version. Specifically, we conduct ablation studies obtained by MetaMask on CIFAR-10 with ResNet-18 and conv encoders in Appendix A.4.8, where ''BT'' denotes Barlow Twins, ''w/o ML'' denotes the ablation model without the proposed $\mathcal{M}$ and the corresponding meta-learning-based learning paradigm, i.e., SimCLR + Barlow Twins, and ''w/o drr'' denotes the ablation model without the dimensional redundancy reduction loss $\mathcal{L_{drr}}$. (a) We conduct experiments to prove the effectiveness of the proposed dimensional mask $\mathcal{M}$ on CIFAR10 with ResNet-18, and the results show that only w/o ML (SimCLR + Barlow Twins) cannot improve the performance of the model by a significant margin, and the proposed approach is crucial to the performance improvement. (b) We conduct experiments on CIFAR10 with ResNet-18 and conv, respectively. The results further prove the effectiveness of the proposed $\mathcal{M}$ and $\mathcal{L_{drr}}$.
>
> Barlow Twins handles dimensional redundancy but suffers dimensional confounder. MetaMask mitigates dimensional confounders by learning and applying a dimensional mask. We conduct experiments to demonstrate our statement, and the results are reported in Figure 10 (Page 24). For the experiments shown in Figure 10 (Page 24) (a), we demonstrate the effectiveness of the proposed dimensional mask $\mathcal{M}$ and the corresponding meta-learning-based training paradigm by directly removing such approaches. The results show that the sole w/o ML only improves Barlow Twins by 0.04, while MetaMask can improve BT by 0.29, which proves the proposed $\mathcal{M}$ is pivotal to the performance promotion. In Figure 10 (Page 24) (b), to verify whether there would be performance gain only from alleviating dimensional confounder without $\mathcal{L_{drr}}$, we evaluate the performance of w/o drr and MetaMask. We observe that for the experiments with ResNet-18, w/o drr (without Barlow Twins) improves SimCLR by 0.2 but cannot reach the performance of BT, and MetaMask can improve both SimCLR and BT by 4.28 and 0.29, respectively. For the experiments with conv, w/o drr improves SimCLR by 1.03 and also outperforms BT by 4.08, and MetaMask improves both SimCLR and BT by 2.27 and 6.22, respectively. Concretely, the performance of w/o drr is related to the performance of SimCLR, and it can always improve SimCLR but underperform the complete MetaMask. MetaMask has consistent best performance by using different encoders.
>
> We consider that $\mathcal{L_{drr}}$ (Barlow Twins loss) could exacerbate dimensional confounder, but as our discussion in Appendix A.4.7 (also A2 in the responses), i.e., dimensions containing confounders are also possible to contain discriminative information, more dimensions with confounders due to $\mathcal{L_{drr}}$ may also carry more discriminative information. Likewise, the model without $\mathcal{L_{drr}}$ may generate representations with over-redundant dimensions so that the total amount of available discriminative information will decrease. However, roughly using the representations with complex dimensional information (without $\mathcal{L_{drr}}$) may result in insufficient discriminative information mining, e.g., w/o ML can only outperform SimCLR by a limited margin. Our proposed MetaMask effectively avoids the appearance of such an undesired phenomenon by leveraging $\mathcal{M}$ and the corresponding meta-learning-based training paradigm, which is supported by the empirical results.

---

> ### Author Response · Authors · 2022-08-09
> **Final response to the reviewer**
>
> Before the end of the discussion phase, we would like to thank the reviewer again for his/her careful review and helpful suggestions.

---

### Official Review · Reviewer_rGkj · 2022-07-11

**Rating:** 6
**Confidence:** 3
**Soundness:** 3 good
**Presentation:** 4 excellent
**Contribution:** 3 good

**Summary:**

This paper handles the problems which existing self-supervised learning algorithms suffer. Authors demonstrate that existing self-supervised learning algorithms suffer dimensional redundancy and dimensional confounder. They propose MetaMask which learns a dimensional mask through meta learning to decrease the learning signals of features having confounders. For the rationales of MetaMask, they provide theoretical support where MetaMask has tighter risk bounds in downstream tasks compared to baselines. They demonstrate that MetaMask combined with competitive self-supervised algorithms has superior performance to baselines on multiple benchmark datasets under two architectures.

**Questions:**

1. (ablation study) Barlow Twins handles dimensional redundancy but suffers dimensional confounder. MetaMask mitigates dimensional confounder by learning and applying a dimensional mask. There would be performance gain only from alleviating dimensional confounder. Can authors show the performance of MetaMask without redundancy-reduction objective function? (I understand that Barlow Twins could exacerbate dimensional confounder.)

**Limitations:**

It would be great if the authors suggest possible issues from the adoption of bi-level optimization such as computational costs.

**Strengths And Weaknesses:**

The paper has several strong and weak points.

Strengths:
1. This paper shows that the existing self-supervised learning algorithms suffer dimensional redundancy and dimensional confounder. The problem is also quite intuitive.
2. Theoretical support for the reasons why MetaMask works is provided. (I have not carefully checked the proofs)
3. Authors conduct extensive experiments and demonstrate that MetaMask combined with existing methods improves performance in most cases.

Weaknesses:
1. MetaMask adopts bi-level optimization, so it would necessarily require significant amounts of additional computational cost during computing second-derivatives. However, authors do not provide any analysis of computational costs.
2. Authors maintain that MetaMask reduces the gradient effect of the dimension containing the confounder (in lines 76-80), but they do not explicitly demonstrate that masked dimensions (having low weights in MetaMask) have confounder (non-discriminative features such as backgrounds). It would be great if the authors suggest experiments about it.
3. The improvement of MetaMask seems marginal in important cases although MetaMask improves baseline performance by combining with them in most cases. Specifically, in Table 2, MetaMask combined with existing algorithms in the modern architecture (ResNet18) shows comparable performance to the most competitive baselines (such as BYOL in CIFAR-10, NNCLR in CIFAR-100, and NNCLR in IN-200).
4. (simple baseline) Applying dropout on $h_i$ in Barlow Twins might be the simple baseline for MetaMask and it would be effective in that BarlowTwins with randomly masked dimensions outperform naive BarlowTwins in Figure 2. Would authors provide the performance of Barlow Twins + dropout?

Miscellaneous minor issues:
1. In Figure 1, (a) and (b) are missing in the figure (only mentioned in the caption).

---

[After reading authors' answers] I appreciate the detailed responses and authors address several of my concerns. Accordingly, I change my rating to weak accept.

---

> ### Author Response · Authors · 2022-08-02
> **Added Experiments and Analyses of Computational Costs, Masking Dimensions that Contains Confounders, Impacts of Encoders’ Ability on MetaMask, Applying Dropouts to Barlow Twins, and A Specifical Ablation Study of MetaMask without $\mathcal{L}_{drr}$ (4/4)**
>
> Q5: In Figure 1, (a) and (b) are missing in the figure (only mentioned in the caption).
>
> A5: Thanks for your scrutiny! We revise this in the rebuttal revised version.
>
> Q6: (ablation study) Barlow Twins handles dimensional redundancy but suffers dimensional confounder. MetaMask mitigates dimensional confounders by learning and applying a dimensional mask. There would be performance gain only from alleviating dimensional confounder. Can authors show the performance of MetaMask without redundancy-reduction objective function? (I understand that Barlow Twins could exacerbate dimensional confounder.)
>
> A6: Thanks for the suggestions! We added the corresponding analysis in Appendix A.4.8 of the rebuttal revised version. Specifically, we conduct ablation studies obtained by MetaMask on CIFAR-10 with ResNet-18 and conv encoders in Appendix A.4.8, where ''BT'' denotes Barlow Twins, ''w/o ML'' denotes the ablation model without the proposed $\mathcal{M}$ and the corresponding meta-learning-based learning paradigm, i.e., SimCLR + Barlow Twins, and ''w/o drr'' denotes the ablation model without the dimensional redundancy reduction loss $\mathcal{L_{drr}}$. (a) We conduct experiments to prove the effectiveness of the proposed dimensional mask $\mathcal{M}$ on CIFAR10 with ResNet-18, and the results show that only w/o ML (SimCLR + Barlow Twins) can not improve the performance of the model by a significant margin, and the proposed approach is crucial to the performance improvement. (b) We conduct experiments on CIFAR10 with ResNet-18 and conv, respectively. The results further prove the effectiveness of the proposed $\mathcal{M}$ and $\mathcal{L_{drr}}$.
>
> Barlow Twins handles dimensional redundancy but suffers dimensional confounder. MetaMask mitigates dimensional confounders by learning and applying a dimensional mask. We conduct experiments to demonstrate our statement, and the results are reported in Figure 10 (Page 24). For the experiments shown in Figure 10 (Page 24) (a), we demonstrate the effectiveness of the proposed dimensional mask $\mathcal{M}$ and the corresponding meta-learning-based training paradigm by directly removing such approaches. The results show that the sole w/o ML only improves Barlow Twins by 0.04, while MetaMask can improve BT by 0.29, which proves the proposed $\mathcal{M}$ is pivotal to the performance promotion. In Figure 10 (Page 24) (b), to verify whether there would be performance gain only from alleviating dimensional confounder without $\mathcal{L_{drr}}$, we evaluate the performance of w/o drr and MetaMask. We observe that for the experiments with ResNet-18, w/o drr (without Barlow Twins) improves SimCLR by 0.2 but cannot reach the performance of BT, and MetaMask can improve both SimCLR and BT by 4.28 and 0.29, respectively. For the experiments with conv, w/o drr improves SimCLR by 1.03 and also outperforms BT by 4.08, and MetaMask improves both SimCLR and BT by 2.27 and 6.22, respectively. Concretely, the performance of w/o drr is related to the performance of SimCLR, and it can always improve SimCLR but underperform the complete MetaMask. MetaMask has consistent best performance by using different encoders.
>
> We consider that $\mathcal{L_{drr}}$ (Barlow Twins loss) could exacerbate dimensional confounder, but as our discussion in Appendix A.4.7 (also A2 in the responses), i.e., dimensions containing confounders are also possible to contain discriminative information, more dimensions with confounders due to $\mathcal{L_{drr}}$ may also carry more discriminative information. Likewise, the model without $\mathcal{L_{drr}}$ may generate representations with over-redundant dimensions so that the total amount of available discriminative information will decrease. However, roughly using the representations with complex dimensional information (without $\mathcal{L_{drr}}$) may result in insufficient discriminative information mining, e.g., w/o ML can only outperform SimCLR by a limited margin. Our proposed MetaMask effectively avoids the appearance of such an undesired phenomenon by leveraging $\mathcal{M}$ and the corresponding meta-learning-based training paradigm, which is supported by the empirical results.
>
> References:
>
> [41] Yifei Wang, Qi Zhang, Yisen Wang, Jiansheng Yang, and Zhouchen Lin. Chaos is a ladder: A new theoretical understanding of contrastive learning via augmentation overlap. arXiv preprint arXiv:2203.13457, 2022.

---

> ### Author Response · Authors · 2022-08-02
> **Added Experiments and Analyses of Computational Costs, Masking Dimensions that Contains Confounders, Impacts of Encoders’ Ability on MetaMask, Applying Dropouts to Barlow Twins, and A Specifical Ablation Study of MetaMask without $\mathcal{L_{drr}}$ (3/4)**
>
> Q3: The improvement of MetaMask seems marginal in important cases although MetaMask improves baseline performance by combining with them in most cases. Specifically, in Table 2 (Page 9), MetaMask combined with existing algorithms in the modern architecture (ResNet18) shows comparable performance to the most competitive baselines (such as BYOL in CIFAR-10, NNCLR in CIFAR-100, and NNCLR in IN-200).
>
> A3: Thanks for the comments. We also observed such a trend: MetaMask improves the benchmark methods with weak encoders by excessively significant margins, but the improvement with strong encoders is relatively limited. Our consideration behind this phenomenon is that although representations learned by both weak and strong encoders (without our method) may contain dimensional confounder, the strong encoder can better capture semantic information so that the dimensional confounder of the learned representation is naturally less, and the useful discriminative information is much more than the representation learned by the weak encoder. [41] provided the theorem and corresponding proof to demonstrate that the contrastive loss can bound the cross-entropy loss in downstream tasks. Therefore, strong encoders better minimize the contrastive loss, which means the representations learned by strong encoders contain more semantic information, and accordingly, fewer dimensional confounders. However, from the results reported in Table 2 (Page 9), our method can still improve the benchmark methods.
>
> We further conduct experiments by imposing random dimensional masks on the learned representations for weak and strong encoders. The results are reported in Appendix A.4.3 of the rebuttal revised version. The comparisons demonstrate that under the consistent setting of the 5% random dimensional mask rate, the results of the weak encoder (conv) range from 52.79 to 54.13, and the results of the strong encoder (ResNet-18) range from 60.71 to 61.06. Note that we conduct 10 trials for each experiment to achieve unbiased results. The quality of the strong encoder's representation is better, and the performance of the strong encoder is more stable (the variance of the strong encoder's results is smaller), which proves that the representation learned by the strong encoder contains less dimensional confounder, and each dimension contains more discriminative information. Therefore, our method improves weak encoders more than strong encoders.
>
> Table 3. The comparisons of Barlow Twins using different dropout ratios on CIFAR10 with ResNet-18.
> | Dropout ratio | Dropout shared between views | Accuracy |
> | :------| :----| :----|
> | 0 | No | 85.72 |
> | 0.01 | No | 85.76 |
> | 0.05 | No | 86.11 |
> | 0.1 | No | 85.72 |
> | 0.2 | No | 85.13 |
> | 0.5 | No | 81.54 |
> | 0.05 | Yes | 31.17 |
> | 0.1 | Yes | 37.15 |
> | 0.5 | Yes | 31.62 |
>
> Q4: (simple baseline) Applying dropout on h_i in Barlow Twins might be the simple baseline for MetaMask and it would be effective in that BarlowTwins with randomly masked dimensions outperform naive BarlowTwins in Figure 2. Would authors provide the performance of Barlow Twins + dropout?
>
> A4: Thank you for this suggestion. We apply dropout to randomly set the features generated by the backbone network to 0 with a given probability. We use two different settings: 1) features from different views are randomly set to 0 independently with 6 probabilities, including 0.01, 0.05, 0.1, 0.2, and 0.5, which is shown as the setting of dropout shared between views is ''No''; 2) for the same sample, we set the same channels of the features from different views to 0. In detail, we let the features from the first view pass the dropout layer, and then record the channels which are set to 0. Finally, we set the same feature channels of the second view to 0 and multiply the features with 1/(1-p), where ''p'' refers to the probability of dropout. In this case, we use three probabilities: 0.05, 0.1, and 0.5. This setting is shown as ''Yes'' for the dropout shared between views. The results are reported in Table 3. We observe that models trained by following the second setting are collapsed, and these models far underperform MetaMask. For the first setting, the performance trend is in accordance with our and the reviewer's expectations, using several well-chosen dropout ratios can improve the performance of Barlow Twins to a certain extent, but the improvement is inconsistent. Furthermore, even the best model of Barlow Twins using the dropout trick cannot achieve comparable performance to MetaMask (as shown in Table 2, Page 9, MetaMask achieves 87.53 on CIFAR10 with ResNet-18). We add this exploration in Appendix A.4.5 of the rebuttal revised version.

---

> ### Author Response · Authors · 2022-08-02
> **Added Experiments and Analyses of Computational Costs, Masking Dimensions that Contains Confounders, Impacts of Encoders’ Ability on MetaMask, Applying Dropouts to Barlow Twins, and A Specifical Ablation Study of MetaMask without $\mathcal{L_{drr}}$ (2/4)**
>
> Q2: Authors maintain that MetaMask reduces the gradient effect of the dimension containing the confounder (in lines 76-80), but they do not explicitly demonstrate that masked dimensions (having low weights in MetaMask) have confounder (non-discriminative features such as backgrounds). It would be great if the authors suggest experiments about it.
>
> A2: Thanks for your suggestion! First, we clarify that MetaMask trains $\mathcal{M}$ by adopting a meta-learning-based training approach, which ensures that $\mathcal{M}$ can partially mask the ''gradient contributions'' of dimensions containing task-irrelevant information and further promote the encoder to focus on learning task-relevant information. So, MetaMask only performs the gradient mask during training instead of physically masking dimensions in the test. We provide theoretical explanation and proofs in Appendix A.1 and A.2. The reasons behind our behavior (adjusting the gradient weight of each dimension in training instead directly masking these dimensions in the test) include: even dimensions that contain dimensional confounders are also possible to contain discriminative information so that lowering the gradient contribution of such dimensions can not only prevent the over-interference of the dimension confounders on the representation learning but also preserve the acquisition of the information of these dimensions. Accordingly, the foundational idea behind self-supervised learning is to learn ''general'' representation that can be generalized to various tasks. In MetaMask, we introduce a meta-learning-based approach to train the dimensional mask $\mathcal{M}$ concerning improving the performance of contrastive learning. However, the theorems, proposed by [41] and us (in Section 5, Appendix A.1, and Appendix A.2), only prove that the contrastive learning objective is associated with the downstream classification task, while there is no evidence to demonstrate the connections between contrastive learning objective and other downstream tasks. Therefore, we did not directly mask the dimensions containing dimensional confounders in the test.
>
> Furthermore, we conduct experiments to explore the performance of the variant that directly masks these dimensions in the test, which is demonstrated in Appendix A.4.7 of the rebuttal revised version. For the exploration of our masking scheme and its variants, we conduct experiments as follows: after training, we collect the final dimensional weight matrix $\mathcal{M}$ and then choose dimensions with weights below average as the masked dimensions. These dimensions are considered to be associated with dimensional confounders. To prove whether these dimensions have confounders, we perform random dimensional masking to these dimensions, and when the masking rate is 100%, the model turns to the variant that directly masks all these dimensions in the test. The experiments are based on SimCLR + MetaMask. Note that we conduct 10 trials per mask rate (except for 0% and 100% mask rates) for fair comparisons. We observe that the original MetaMask (i.e., mask rate is 0%) achieves the best performance on average, and MetaMask outperforms the variant masking the dimensions with confounders by a significant margin, which proves that our proposed approach, i.e., masking the ''gradient contributions'' of dimensions in the training is more effective than the compared approach, i.e., directly masking dimensions in the test. While, several trials with specific mask rates demonstrate better performance than MetaMask, which proves that the dimensions filtered by MetaMask indeed contain dimensional confounders. Additionally, observing the results reported in Figure 2 (Page 3) and the results in Appendix A.4.7, we find that the results achieved by the proposed variants are better than Barlow Twins with random dimensional masks on average, which can further prove the filtered dimensions containing confounders and that MetaMask indeed assigns lower gradient weights to the dimensions containing confounders.

---

> ### Author Response · Authors · 2022-08-02
> **Added Experiments and Analyses of Computational Costs, Masking Dimensions that Contains Confounders, Impacts of Encoders’ Ability on MetaMask, Applying Dropouts to Barlow Twins, and A Specifical Ablation Study of MetaMask without $\mathcal{L_{drr}}$ (1/4)**
>
> We thank the reviewer for the valuable comments and constructive suggestions. We are encouraged that the reviewer found that this work is novel and technically sound, and the presentation is excellent. The mentioned issues are addressed as follows:
>
> Table 1. The complexity comparisons between MetaMask and benchmark methods on the CIFAR10 dataset. Note that for fair comparisons, this experiment is based on 1 GPU of NVIDIA Tesla V100.
> |Methods | Parameters | Training time cost for an epoch |
> | :------| :----| :----|
> | ResNet-18 | 11.2M | - |
> | SimCLR | 13M | 70s |
> | Barlow Twins | 22.7M | 80s |
> | SimCLR + Barlow Twins | 24.6M | 85s |
> | MetaMask | 24.6M + 512 | 210s |
>
> Table 2. The comparisons between MetaMask and benchmark methods on the CIFAR10 dataset by using the same total time costs. Note that for fair comparisons, this experiment is based on 1 GPU of NVIDIA Tesla V100.
> |Methods | Epoch | Training time cost | Accuracy |
> | :------| :----| :----| :----|
> | SimCLR | 2400 | 46h | 81.75 |
> | Barlow Twins | 2100 | 46h | 85.71 |
> | SimCLR + Barlow Twins | 2000 | 47h | 85.79 |
> | MetaMask |800 | 46h | 86.01 |
>
> Q1: MetaMask adopts bi-level optimization, so it would necessarily require significant amounts of additional computational cost during computing second-derivatives. However, authors do not provide any analysis of computational costs.
>
> A1: To compare the training complexity of MetaMask and benchmark methods, we conduct experiments on CIFAR10 by using ResNet-18 as the backbone network. The results are demonstrated in Table 1, which shows that the number of parameters used by MetaMask is close to the ablation model, i.e., SimCLR + Barlow Twins, and Barlow Twins. Compared with Barlow Twins, SimCLR + Barlow Twins only adds a MLP with several layers as the projection head for the contrasting learning of SimCLR. Comparing SimCLR + Barlow Twins and MetaMask, we only add a learnable dimensional mask $\mathcal{M}$ in the network of MetaMask. Note that to decrease the time complexity of MetaMask, we create an additional parameter space to save the temporary parameters for computing second-derivatives, but such cloned parameters do not participate in the calculation of the network during training.
>
> For the time complexity, due to the learning paradigm of meta-learning (second-derivatives technique), MetaMask has larger time complexity than benchmark methods (including the ablation model SimCLR + Barlow Twins). While, we further conduct experiments to evaluate the performance of the compared methods using similar total training time costs. The results are reported in Table 2, which demonstrates that MetaMask still achieves the best performance, and MetaMask can also beat the ablation model SimCLR + Barlow Twins. This proves that although the time complexity of MetaMask is a little bit high, the improvement of MetaMask is consistent and solid. We also include these results and the corresponding analysis in Appendix A.4.4 of the rebuttal revised version. Thank you for this suggestion.

---

> ### Author Response · Authors · 2022-08-09
> **Response to the follow-up**
>
> We thank the reviewer for his/her efforts and time to review our paper. The comments and suggestions are very professional and constructive.

---

### Official Review · Reviewer_m2DS · 2022-07-12

**Rating:** 8
**Confidence:** 4
**Soundness:** 4 excellent
**Presentation:** 4 excellent
**Contribution:** 3 good

**Summary:**

This paper first investigates the problem of dimensional confounder in contrastive learning, which refers to a subset of dimensions learning only task-irrelevant background information. The author shows the existence of the dimension confounder, where the performance of Barlow Twin starts to drop and eventually collapses after the dimension of the projection head reaches a certain point. The author then proposes to learn a dimensional mask approach, MetaMask, consisting of learnable weights to reweight each dimension of the encoder output in contrastive learning, so that the dimensional confounder can be assigned lower weights. The authors show theoretically that the proposed MetaMask achieves tighter risk bounds (lower and upper) for downstream classification tasks, and empirically improves the performances on downstream tasks and robustness towards dimension sizes by applying MetaMask to the prior models.

**Questions:**

Q1: Figure 1(a) is interesting -- the proposed MetaMask seems to have much higher contributions on most dimensions than SimCLR -- why is this and is this desirable?

Q2: How the experiments for Figure 1(b) are a little unclear. How many layers were you using? Were you using the same dimension for all the layers of the projection head, or did you shrink the dimension at the last layer to serve as a dimension bottleneck? These are important because your results sharply contrast with Barlow Twins'.

Q3: Could you provide the dimensional mask rate that improves the performance on both ImageNet and CIFAR-10? Although you claim the ImageNet rate is higher, the difference seems indistinguishable from Figure 2.

Q4: The claim you made, that (Line 50) "the dimensionality of the projection head ... acts as a dimensionality bottleneck" is not the same as the claim from Barlow Twin, "the output of the ResNet is kept fixed to 2048, which acts as a dimensionality bottleneck". The bottleneck, according to Barlow Twin, seems to be the ResNet output dimension, rather than the projection head dimension. Could you clarify this?

**Limitations:**

The authors discussed the limitations of the work. The limitation states that the proposed MetaMask is specific towards improving contrastive learning but could not yet show the theoretical guarantees of MetaMask on other self-supervised learning methods, such as masked image modeling. This is a correct assertion and the discussions of limitations are helpful for the community.

The authors have not shed light on the negative societal impact.

**Strengths And Weaknesses:**

Originality:

Strengths: the idea is novel, addressing the issue of dimensional confounders by reweighting encoder outputs using meta-learning techniques.  The idea is related to MAXL[1], but the two approaches are different. MAXL applies a mask with weights on the softmax logits in a hierarchical binary prediction setup, where the logits are single scalars. On the other hand, this work applies a mask with weights on the encoder output for contrastive learning, where the output has a large dimension.

The reviewer is not familiar with meta-learning literature and only evaluates the originality from a contrastive learning perspective.

Quality:

Strengths: the idea overall has technical soundness, with theoretical justification on the proposed MetaMask creating tighter lower and upper bounds of risk, and shows reducing the risk can improve downstream performances.

Weaknesses: The reviewers have some additional clarification questions listed in the next section.

The reviewer cannot verify the correctness of Equation 8.

Clarity:

Strengths: the paper is very well written, with clear motivation, thorough related work, a succinct method section, and well-organized theoretical analysis and experimental results.

Significance:

Strengths: the community can adopt the method to reduce the impact of the dimensional confounder, especially in large models. The performance improvements are significant in some cases (Table 2, SimCLR on STL-10 and NNCLR on CIFAR-10).

[1] Liu, Shikun, Andrew Davison, and Edward Johns. "Self-supervised generalisation with meta auxiliary learning." Advances in Neural Information Processing Systems 32 (2019).

---

> ### Author Response · Authors · 2022-08-02
> **Clarifications of Theoretical Proof, Analyses and Descriptions of Experiments in Figure 1 and Figure 2, and Discussions on the Bottleneck Statement and Negative Societal Impact (2/2)**
>
> Q4: Could you provide the dimensional mask rate that improves the performance on both ImageNet and CIFAR-10? Although you claim the ImageNet rate is higher, the difference seems indistinguishable from Figure 2.
>
> A4: We would like to clarify that the experiments conducted in Figure 2 (Page 3) are based on random masks, which aim to prove the existence of dimensional confounders, i.e., if the dimensional confounder is randomly masked, the performance will be promoted. The results support our statement. For the dimensional mask rate that improves the performance on both ImageNet and CIFAR-10, we observe from the experiments and find that around 0.5% may improve the performance on both datasets. The reasons behind using the 0.5% mask rate can improve the performance includes: 1) the encoder is strong (ResNet) so that most of the dimensions contain discriminative information; 2) the masking scheme is a random process so that it is hard to exactly mask the dimensions containing confounders.
>
> Thanks for your careful review! This is a typo of the statement, and such a claim is contrary to the following explanation from Line 67 to Line 73, and also this typo claim is contrary to the experiments in Figure 2 (Page 3). The dimension-to-class ratio on CIFAR-10 (51.2) is much higher than that on ImageNet (2.048). Therefore, the presence of dimensional confounder naturally remains at a low level on ImageNet since supporting $1000$-category classification requires a large amount of heterogeneous discriminative information. Thus, the dimensional mask rate should be lower on ImageNet so that the performance of the model can be improved. We have revised this typo in the rebuttal revised version.
>
> Q5: The claim you made, that (Line 50) "the dimensionality of the projection head ... acts as a dimensionality bottleneck" is not the same as the claim from Barlow Twin, "the output of the ResNet is kept fixed to 2048, which acts as a dimensionality bottleneck". The bottleneck, according to Barlow Twin, seems to be the ResNet output dimension, rather than the projection head dimension. Could you clarify this?
>
> A5: Thanks for pointing out this. We consider that the statement in our paper is similar to the statement in Barlow Twins to a certain extent. Specifically, our statement describes the dimensionality bottleneck only in the training phase, and the features are generated by the cascading structure of the backbone and projection head, and thus the projection head with fixed output dimensionality can be treated as the dimensionality bottleneck for the features. The statement in Barlow Twins describes the dimensionality bottleneck in both training and inference phases. In the inference phase, the projection head is discarded, and the representations are only generated by the backbone. Thus, the backbone can be treated as the dimensionality bottleneck for the representation. The intuition behind both statements is shared, i.e., the fixed dimensionality of the network can restrict the dimensionality of the generated representation/feature.
>
> Concretely, we thank the reviewer again, and we will change the statement from the current version to the statement in Barlow Twins' original paper, because that statement is based on the understanding of Barlow Twins's behavior. We believe it is better to restate the original claim.
>
> Q6: The authors have not shed light on the negative societal impact.
>
> A6: Thanks for mentioning this. Because this work presents a general method to tackle the dimensional confounder issue for self-supervised contrastive learning, we did not see particular foreseeable negative societal impacts and ethical consequences.
>
> References:
>
> [14] Jure Zbontar, Li Jing, Ishan Misra, Yann LeCun, and Stéphane Deny. Barlow twins: Self-supervised learning via redundancy reduction. In Proceedings of the 38th International Conference on Machine Learning, ICML 2021, 18-24 July 2021, Virtual Event, Proceedings of
> Machine Learning Research. PMLR, 2021.
>
> [41] Yifei Wang, Qi Zhang, Yisen Wang, Jiansheng Yang, and Zhouchen Lin. Chaos is a ladder: A new theoretical understanding of contrastive learning via augmentation overlap. arXiv preprint arXiv:2203.13457, 2022.

---

> ### Author Response · Authors · 2022-08-02
> **Clarifications of Theoretical Proof, Analyses and Descriptions of Experiments in Figure 1 and Figure 2, and Discussions on the Bottleneck Statement and Negative Societal Impact (1/2)**
>
> We appreciate the thoughtful feedback of the reviewer m2DS. We are glad the reviewer found that this work is novel and technically sound, the writing is clear. The mentioned issues are addressed as follows:
>
> Q1: The correctness of Equation 8.
>
> A1: Thanks for the careful review! Theorem 4.2 in [41] provides sufficient proof to the statement that the contrastive loss in the self-supervised learning stage can constrain the upper and lower bounds of the cross-entropy loss in the supervised learning stage for downstream tasks. In Theorem 5.1 (Page 7), we extend this theorem to fit the learning paradigm of MetaMask, i.e., adding the learnable dimensional mask into the theorem. We have checked our proof and did not find flaws/errors.
>
> Q2: Figure 1(a) is interesting -- the proposed MetaMask seems to have much higher contributions on most dimensions than SimCLR -- why is this and is this desirable?
>
> A2: First, we clarify that the abscissa axis represents the feature dimensions, and the ordinate axis represents samples of different classes in Figure 1(a) (Page 2). The observation shows that compared with SimCLR, our method using redundancy-reduction can indeed learn representations with decoupled dimensions. For the classification contributions, we further find that some dimensions of the representation learned by MetaMask indeed have higher contributions, while the difference in the classification contributions of the representation learned by SimCLR is relatively small. We consider the reason behind such a result is that the redundancy-reduction technique, proposed by [14], empowers our method to learn dimension-decoupled representation, while the methods without such a regularization, e.g., SimCLR, may learn a representation that has many redundant dimensions, i.e., many dimensions contain very similar information. Therefore, for our method, the dimensions containing discriminative information naturally have a higher classification contribution. For SimCLR, multiple dimensions contain the shared information, and the differences are not very large so that the contribution of each individual dimension is weakened.
>
> The foundational idea of self-supervised learning is to learn a general representation that can be generalized in various downstream tasks. The representation learned by SimCLR cannot sufficiently explore the semantic information from the input because of the dimensional redundancy, while the representation learned by our method can learn dimension-decoupled representation so that more semantic information can be contained by the representation if the dimensionality of the projection head is suitable.
>
> Q3: How the experiments for Figure 1(b) are a little unclear. How many layers were you using? Were you using the same dimension for all the layers of the projection head, or did you shrink the dimension at the last layer to serve as a dimension bottleneck? These are important because your results sharply contrast with Barlow Twins'.
>
> A3: We follow the official code of Barlow Twins. Specifically, in the experiments of Figure 1(b) (Page 2), three layers are used. We use the same dimension for the first two layers of the projection head, and only change the dimension of the last layer, which is as same as the implementation in the paper of Barlow Twins (Section 4 in [14]). For the experiments of CIFAR10, the dimensions of the first two layers are both 2048, and the dimension of the last layer is changing from 512 to 12288. For the experiments of ImageNet, the dimensions of the first two layers are 8192, and the dimension of the last layer ranges from 512 to 40960.
>
> We think the reasons why our results sharply contrast with Barlow Twins are: 1) Barlow Twins does not take experiments on small-scale datasets, e.g., CIFAR10; 2) the dimensionality taken by Barlow Twins is not large enough. In particular, the projection head dimensionality range of Barlow Twins is 16 to 16384, while that in our paper is 512 to 40960, and the results of lower dimensionality are consistent between Barlow Twins and ours.
>
> Furthermore, in further exploration, we propose that such a model collapsing phenomenon is due to two reasons: 1) the proposed dimensional confounder; 2) the curse of dimensionality. And we elaborate on the difference and our intuitive analysis in Appendix A.3. The experiments, shown in Figure 4 (Page 9) and Figure 7 (Page 19), demonstrate that MetaMask can yield significant performance boosts against the mode collapse. We have added the experimental details and explanations on the results between Barlow Twins and our method in Section 6 and Appendix A.4 of the rebuttal revised version. Please refer to the codes in the supplementary file for the detailed implementation.

---

> ### Comment · Reviewer_m2DS · 2022-08-09
> **Follow-up**
>
> I thank the author to provide thorough clarifications of all the questions I have. I am increasing the score by one point, and recommend the acceptance of this work. I also recommend the authors incorporate the answers provided in this rebuttal to an updated draft, especially the answers to Q3, which provides the details of the experiment that serves as the main motivation of this paper, and Q5, which contrasts with the observation from Barlow Twins because of the scale of dimensions.

---

> > ### Author Response · Authors · 2022-08-09
> > **Response to the follow-up**
> >
> > We are glad to thank the reviewer for the constructive comments and suggestions, and we will add the important rebuttals to the updated draft, including the answers to Q3 and Q5.

---

### Author Response · Authors · 2022-08-08
**Gentle Reminder: Follow-up**

Dear Reviewers,

Thank you again for your time and effort in reviewing our paper.

In our early response, we have included detailed analyses and descriptions of the proposed method. Moreover, we have conducted multiple experiments prove the effectiveness of the proposed method.

Since the discussion stage is closing soon, we would be grateful if you could let us know whether our responses and revised manuscript have addressed your concerns and whether there are further comments.

Sincerely, Authors

---

### Meta-Review · Area_Chair_RFWY · 2022-08-23

**Recommendation:** Accept
**Confidence:** Certain

**Metareview:**

This paper starts with the experimentally findings that the interference of task irrelevant information and the disadvantages of sample inefficiency in contrastive learning appear due to dimensional redundancy and dimensional confounder. Based on these experimental findings, the authors propose a dimensional mask learning method based on bi-level optimization. Finally, the theoretical basis for learning through meta mask is also presented. All reviewers agreed with the strength of the proposed method. There were some concerns such as insufficient ablation study, but they were all resolved through the authors' rebuttal and subsequent discussion. I hope the authors make it sure that the concerns raised in the review process can be resolved in the final version.

**Award:**

No

---

### Decision · Program_Chairs · 2022-09-14

Accept